# Prelimbic cortex to ventral tegmental area projection regulates early social isolation stress-potentiated heroin seeking in mice

Yunwanbin Wang [1,4], Shuwen Yue[1,4], Fengwei Yang[2], Lu Chen[1], Archana Singh[1], Magnus Marciniak[1], Wei Wei[1] & Zi-Jun Wang [1,3] ✉

Early-life adversities increase vulnerability to substance use disorders, which are characterized by persistent, uncontrollable drive to seek drugs, often leading to relapse. Previously, we reported that early social isolation (ESI) during adolescence potentiates heroin-seeking in mice. However, the underlying neurobiology remains unknown. Here, we found that ESI aggravated heroin-induced neuronal dysfunction in prelimbic cortex (PrL) to ventral tegmental area (VTA) projecting neurons. Activating PrL->VTA projection attenuated ESI-potentiated heroin seeking, alongside normalized neuronal function. RNA-seq revealed that ESI and heroin convergently altered genes regulating morphogenesis and metabolism, with *Tmsb4x* (thymosin β4) as a key gene. ESI and heroin interaction affected genes regulating cell cycle and DNA damage response, with *Mcm3* and *Mcm7* (minichromosome maintenance proteins 3/7) as hubs. PrL thymosin β4 infusion or CRISPR-Cas9-mediated PrL->VTA projection-specific *Mcm3/7* knockdown attenuated ESI-potentiated heroin-seeking and neuronal hypofunction. Our study suggests that ESI-potentiated heroin relapse is associated with neuronal and transcriptional alterations in PrL->VTA projection.

Opioid use disorder (OUD) is a chronic, life-long disease characterized by compulsive drug taking and persistent vulnerability to relapse. OUD and opioid addiction continue to affect millions of people in the US (about 3 million) and worldwide (about 16 million), including more than 500,000 people in the US who are dependent on heroin[1]. The high relapse rate is one of the most difficult challenges in treating OUD[2]. Stress has been reported to induce vulnerability to opioid relapse[3–5] by affecting the gene expression in key brain regions within the reward pathway. However, the specific brain circuits contributing to stress-induced relapse vulnerability, as well as the corresponding molecular changes within key brain circuits, remain unclear.

Adolescence is a critical period when behavioral and brain circuitry maturation occurs[6–9]. Stress during this key period results in higher rates of psychiatric disorders, including drug addiction. Human

studies indicate that chronic stress during adolescence (or early life adversity) can cause persistent maladaptations that not only increase addiction risk[10–12] but also increase relapse vulnerability[4,13,14]. Therefore, it is important to understand the underlying mechanisms by which early life stress impacts relapse.

Social interaction during early life plays an essential role in cognitive development. Human studies show that individuals who experience social isolation stress during early life have a higher risk of experiencing mental disorders co-occurring with addiction, including anxiety disorder, social withdrawal, depression, and schizophrenia[15–18]. Preclinical studies report that social isolation during adolescence is associated with increased drug taking[19] and drug seeking[20,21]. Using the mouse self-administration (SA) model, we previously reported that early social isolation (ESI) stress potentiates heroin-seeking behavior

[1]Department of Pharmacology & Toxicology, School of Pharmacy, University of Kansas, Lawrence, KS, USA. [2]Department of Biostatistics and Data Science, University of Kansas Medical Center, Kansas City, KS, USA. [3]Cofrin Logan Center for Addiction Research and Treatment, University of Kansas, Lawrence, KS, USA. [4]These authors contributed equally: Yunwanbin Wang, Shuwen Yue. ✉e-mail: zjwang@ku.edu

after forced abstinence[21]. Yet, the underlying mechanisms remain elusive.

The medial prefrontal cortex (PFC), which sends glutamatergic innervation to multiple subcortical regions within the mesolimbic dopamine (DA) pathway, is involved in heroin seeking[22–27]. Opioids induce PFC dysfunction[27–31], which may disrupt these connections and lead to relapse. Interestingly, social experience during adolescence plays a key role in the functional development of brain circuitry, and social deprivation during this time induces long-lasting malfunction in the mPFC and its projecting areas[32–35]. Therefore, it is possible that ESI-potentiated heroin seeking is mediated by functional alterations in mPFC projections.

The medial PFC is topographically organized into various sub-areas, such as the prelimbic (PrL) and the infralimbic (IL) cortex. Interestingly, the role of PrL and IL regions in heroin seeking is inconsistent throughout the literature (e.g., inactivation of PFC enhances cue-induced heroin reinstatement[36]; other studies have found an attenuation of cue-, heroin-, or context-induced heroin reinstatement[23–25]). These discrepancies may, at least partially, be due to the distinct roles of different PFC efferents. In fact, there are diverse populations of projecting neurons within layer V of the PFC[37,38] projecting to the ventral tegmental area (VTA), nucleus accumbens (NAc), or amygdala. These projections have distinct molecular signatures and play different roles in the regulation of motivational behaviors[35,37,38].

The PrL region sends projections to the VTA[39,40], and this projection may contribute to the PFC-gated regulation of VTA neuronal activity and the extracellular DA levels within the mesolimbic reward pathway[41–48]. Studies have shown that inactivation of the PrL to the rostromedial tegmental nucleus projection potentiates cue-induced cocaine seeking[49]. However, the role of PrL to VTA projection in heroin seeking is still unknown. Additionally, our previous studies showed that ESI stress enhances heroin-seeking behaviors and alters c-Fos expression in both PrL and VTA[21]. Therefore, in this study, we examined the role of PrL to VTA (PrL->VTA) projections in heroin seeking in both group-housed and socially isolated mice.

To do so, we used the forced abstinence model and focused on the PrL rather than the IL, which is involved in extinction learning[50]. Using whole-cell patch-clamp recording, we identified reduced neuronal firing and excitatory synaptic transmission in PrL->VTA projections after ESI stress and heroin exposure. Chemogenetic activation of PrL->VTA projections alleviated ESI-potentiated heroin seeking and restored neuronal function in this circuit. Finally, using translating ribosome affinity purification (TRAP)-coupled with RNA-seq, we found the transcriptional alterations within this projection that drive early life stress-potentiated heroin seeking, with *Tmsb4x* (encoding thymosin β4), *Mcm3*, and *Mcm7* (encoding minichromosome maintenance proteins 3 and 7) identified as hub genes. Further manipulation—via thymosin β4 infusions in the PrL or CRISPR-Cas9-mediated conditional knockdown of *Mcm3* and *Mcm7* in the PrL->VTA projection—validated their key roles in mediating ESI-induced susceptibility to heroin.

## Results

### ESI stress potentiates heroin abstinence-induced neuronal dysfunction in PrL->VTA projecting neurons

Our previous study showed that ESI stress potentiates heroin seeking after prolonged abstinence and alters neuronal activity in PrL and VTA[21]. Therefore, we first investigated whether PrL->VTA projecting neurons showed abnormal neuronal function after heroin abstinence and ESI stress exposure. As previously reported[21,35], C57BL/6J mice were placed in standard group housing (GH) or chronic ESI (5 weeks) housing conditions right after weaning. At the age of 8 weeks, GH and ESI mice received water SA training followed by jugular catheterization. Then they were subjected to saline or heroin SA (0.05 mg/kg/infusion) for 10 days (Fig. 1A). Both GH and ESI mice had substantially higher infusion numbers (Fig. 1B) and total active responses (Fig. S1A) for heroin

compared to saline. Moreover, the total active responses for heroin increased throughout the 10 training sessions (Fig. S1A), indicating an increased behavioral response to heroin. Additionally, no significant changes were found between GH and ESI mice in heroin or saline infusions (Fig. 1B) as well as total active responses (Fig. S1A). These data suggest that both GH and ESI mice self-administered significantly more heroin than saline, and there was no pre-existing difference between GH and ESI mice in heroin intake under the dose we used.

Then, mice were subjected to forced abstinence after SA training, during which time retrograde CTB 488 was injected into the VTA (Fig. 1C). After 14 days of abstinence, mice were euthanized for whole-cell patch-clamp recording, and PrL->VTA projecting neurons were visualized under a fluorescent microscope (Fig. 1C). As neuronal activity and excitability were required to maintain neuronal function, we first recorded synaptic-driven spontaneous action potential (sAP) using modified artificial cerebrospinal fluid (ACSF) with low $Mg^{2+}$ to elevate brain slice activity. In general, the recorded firing frequency for green fluorescence-positive cells (PrL pyramidal neurons projecting to the VTA) was less than 3 Hz, which is consistent with previous publications[51,52]. Furthermore, we found that there were significant main effects of heroin abstinence and ESI stress on the frequency of sAP (Fig. 1D). Specifically, the frequency of sAP was significantly lower in the PrL->VTA projecting neurons from mice that underwent heroin abstinence compared to mice from the saline group (Fig. 1D, GH HER vs GH SAL, $p < 0.05$). In addition, sAP frequency was notably lower in the PrL->VTA projecting neurons from ESI mice compared to GH mice (Fig. 1D, ESI SAL vs GH SAL, $p < 0.05$). Moreover, the inhibition of sAP frequency was potentiated by the interaction of heroin abstinence and ESI stress (Fig. 1D, ESI HER vs GH HER, $p < 0.05$; ESI HER vs ESI SAL, $p < 0.05$). Interestingly, there was no change in the sAP amplitude (Fig. S1C), half-width (Fig. S1D), rise time (Fig. S1E), and decay time (Fig. S1F) in the PrL->VTA projecting neurons across all groups, suggesting that the change in firing frequency after heroin abstinence was not attributed to the altered action potential (AP) amplitude and kinetics[53].

Next, we explored whether decreased neuronal firing was related to changes in intrinsic excitability. To do so, we recorded APs elicited by injecting a series of depolarizing currents (eAP). We found that the frequency of evoked spikes was unaltered by either heroin abstinence or ESI stress alone (Fig. 1E); however, the interaction effect (current × drug × stress) was significant (Fig. 1E, $F_{1, 68\ (intercept)} = 872.303$, $p < 0.001$, multi-way ANOVA). Specifically, the frequency of evoked spikes was significantly lower in ESI mice that underwent heroin abstinence compared to GH mice that underwent saline abstinence (Fig. 1E, ESI HER vs GH SAL, $p < 0.05$ at injected current of 140, 160, and 180 pA, $p < 0.01$ at injected current of 200 pA).

Excitatory synaptic transmission is a key component to drive neuronal firing, and abnormal excitatory synaptic transmission has been implicated in opioid-[27,31] and isolation stress-[32,54,55] induced neuronal dysfunction. Therefore, we measured changes in spontaneous excitatory postsynaptic currents (sEPSC) in PrL->VTA projecting neurons from GH and ESI mice that underwent saline or heroin SA followed by abstinence. As shown in Fig. 1F, G, either heroin or ESI stress alone had a significant effect on the frequency of sEPSC. In particular, sEPSC frequencies were inhibited by ESI stress (ESI SAL vs GH SAL, $p < 0.001$) or heroin abstinence (GH HER vs GH SAL, $p < 0.05$). Heroin-induced reduction of sEPSC frequencies was also potentiated by ESI stress (ESI HER vs GH HER, $p < 0.05$). The amplitude of sEPSC was not altered across all groups (Figs. 1H and S1G).

All the data above suggest that heroin abstinence and ESI stress can induce hypofunction of PrL->VTA projecting neurons, which can be potentiated by the interaction of heroin and ESI stress. Moreover, the hypofunction induced by this interaction may be linked to decreased intrinsic excitability and excitatory synaptic transmission in PrL->VTA projecting neurons.

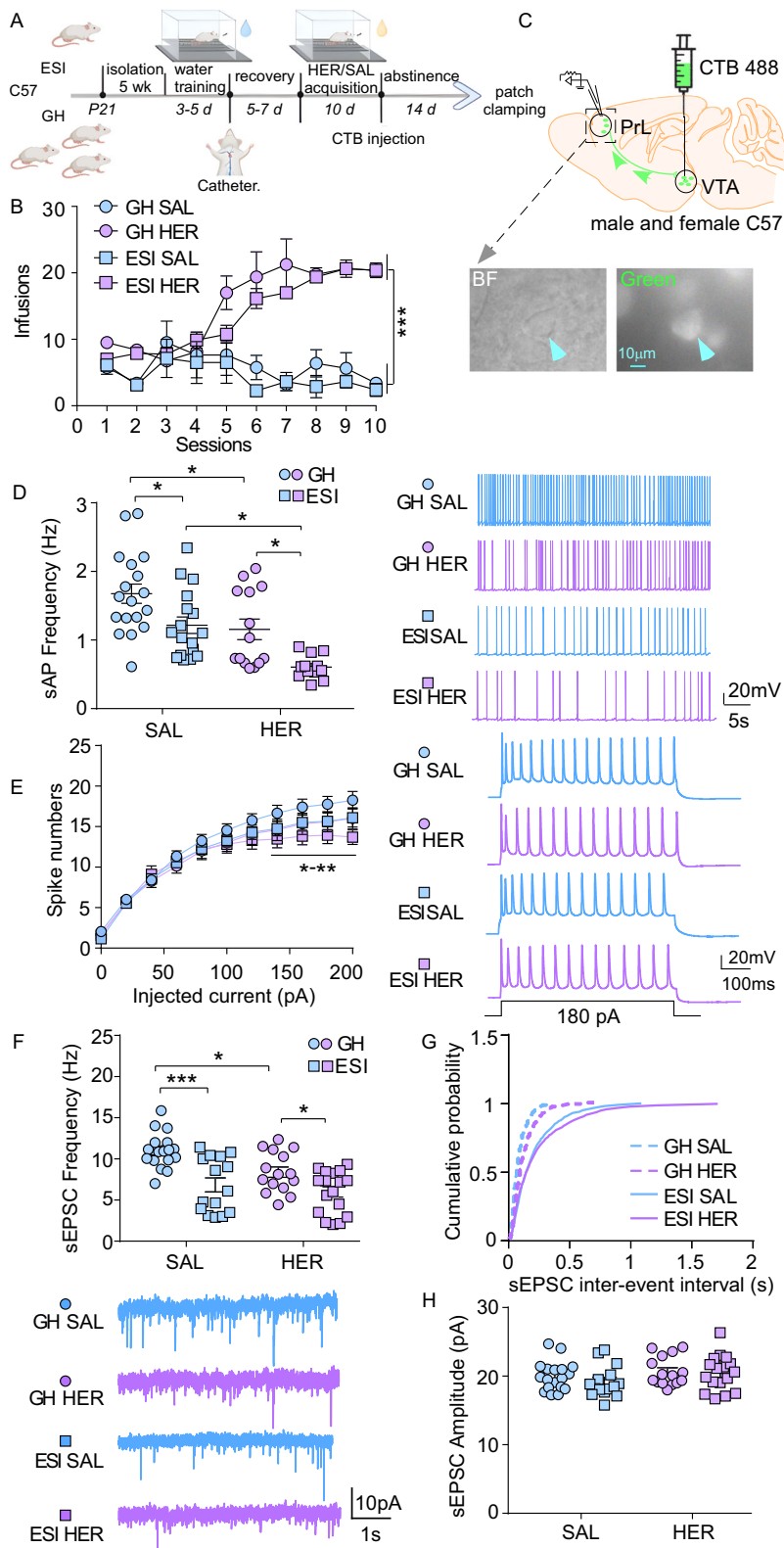

## Activation of PrL->VTA projecting neurons attenuates ESI-potentiated heroin seeking

We next sought to explore whether activation of the PrL->VTA projection can attenuate ESI-potentiated heroin-seeking behavior (Fig. 2A). To do so, a chemogenetic tool, DREADDs (designer receptor exclusively activated by designer drugs), was used. A mCherry-tagged retrograde double-floxed Gq (hM3D(Gq)) virus was injected bilaterally into the VTA of C57BL/6J mice, and the GFP-tagged Cre virus was injected into the PrL to initiate the expression of hM3Dq receptors on the PrL->VTA projecting neurons (Fig. 2B, C). sAP frequency of PrL->VTA projecting neurons was recorded by whole-cell patch-clamp (Fig. S2A, B). Neurons within the PrL->VTA projection were selected by double labeling of GFP and mCherry under a microscope (Fig. S2A). Upon C21 bath application (10 μM[56]), PrL->VTA projecting neuron

**Fig. 1 | Early social isolation (ESI) stress potentiates hypofunction in PrL->VTA projecting neurons during heroin abstinence. A** Experimental timeline for water training, saline (SAL) or heroin (HER) self-administration (0.05 mg/kg/infusion for HER), cholera toxin subunit B (CTB) 488 injection (300 nL/hemisphere), and electrophysiology in C57BL/6 mice. **B** Mean numbers of SAL or HER infusions per session during the acquisition of SA ($n = 8$ mice/group, multi-way ANOVA, $F_{1, 28 \text{ (drug)}} = 67.32$, $p < 0.001$. GH SAL group-housed saline, GH HER group-housed heroin, ESI SAL early social isolated saline, ESI HER early social isolated heroin). **C** Schematic of CTB 488 injection site in the VTA and visualization of PrL->VTA projecting neurons in the PrL during patch-clamp recording. PrL prelimbic area, VTA ventral tegmental area. **D** Left: Graph showing firing frequency of spontaneous action potentials (sAP) in PrL->VTA projection across all groups following abstinence ($n = 18$ [GH SAL], 17 [ESI SAL], 14 [GH HER] and 12 [ESI HER] cells from 4 mice/group, two-way ANOVA, $F_{1, 57 \text{ (drug)}} = 18.93$, $p < 0.0001$, $F_{1, 57 \text{ (stress)}} = 15.04$,

$p = 0.0003$). Right: representative sAP traces. **E** Left: Number of spikes elicited by current steps ($n = 20$ [GH SAL], 20 [ESI SAL], 15 [GH HER] and 17 [ESI HER] cells from 4 mice/group, multi-way ANOVA, $F_{1, 68 \text{ (intercept)}} = 872.3$, $p < 0.001$, $F_{10, 680 \text{ (current)}} = 434.8$, $p < 0.001$). Right: Examples of sAP with current injections at 180 pA. **F** Top: Bar graph showing the frequencies of spontaneous excitatory postsynaptic currents (sEPSCs) in PrL->VTA neurons ($n = 18$ [GH SAL], 15 [ESI SAL], 15 [GH HER] and 17 [ESI HER] cells from 4 mice/group, two-way ANOVA, $F_{1, 61 \text{ (drug)}} = 7.7$, $p = 0.0073$, $F_{1, 61 \text{ (stress)}} = 26.8$, $p < 0.0001$). Bottom: Representative sEPSC traces. **G** Cumulative distribution of the sEPSCs inter-event intervals. **H** Bar graph showing the amplitudes of sEPSCs in PrL->VTA neurons ($n = 18$ [GH SAL], 15 [ESI SAL], 15 [GH HER], and 17 [ESI HER] cells from 4 mice/group). Data are expressed as mean ± SEM, *$p < 0.05$, **$p < 0.01$, ***$p < 0.001$. **A** created in BioRender. Wang, Z. (2025) https://BioRender.com/6kfkeg5. Source data are provided as a Source data file.

firing rate was significantly increased (Fig. S2B). We also showed that mice receiving C21 injection (1 mg/kg, intraperitoneal[57]) expressed more c-Fos positive cells in PrL compared to the vehicle group (Fig. S2C–F). These data suggest that hM3D(Gq) can selectively activate neurons projecting from the PrL to VTA.

Then, GH and ESI C57BL/6J mice were subjected to heroin SA (Figs. 2D and S3A, B) followed by forced abstinence, during which time they received double-viral injection (DIO-hM3Dq-mCherry-AAVrg into the VTA and Cre-GFP into the PrL, Fig. 2A–C). After virus expression, mice were counterbalanced based on their behavioral performance (the infusion numbers during SA, Fig. 2D) and assigned to receive vehicle (Veh) or C21 injection (1 mg/kg, i.p.[57]) 30 min before the heroin-seeking test. Consistent with previous results showing ESI stress potentiates heroin seeking[21], we found that the total active responses (Fig. 2E, ESI Veh vs GH Veh, $p < 0.001$) and total responses (Fig. S3C, ESI Veh vs GH Veh, $p < 0.01$) during the heroin-seeking test were significantly higher in vehicle ESI mice compared to vehicle GH mice. Chemogenetic activation of the PrL->VTA projection with C21 was able to lower the total active responses (Fig. 2E, ESI C21 vs ESI Veh, $p < 0.001$) and total responses (Fig. S3C, ESI C21 vs ESI Veh, $p < 0.001$) in ESI mice. Interestingly, the activation of the PrL->VTA projection also attenuated total active responses (Fig. 2E, GH C21 vs GH Veh, $p < 0.01$) and total responses (Fig. S3C, GH C21 vs GH Veh, $p < 0.001$) in GH mice. Notably, although C21 injection slightly reduced total inactive responses in GH mice (but not ESI mice) during the heroin-seeking test (Fig. 2F), the general locomotion (Fig. 2G) was completely intact, suggesting the attenuation of heroin-seeking behavior was not due to changes in general locomotor activity. These changes in heroin-seeking behavior were neither due to pre-existing bias in heroin intake, as the infusion numbers (Fig. 2D), total active and inactive responses (Fig. S3A, B) during heroin SA training did not change across groups.

To evaluate any C21 off-target behavioral effects, chemogenetic experiments were performed with a DREADD-free control group receiving the same dose of C21 injection (1 mg/kg, i.p.). Similarly, after 10 days of heroin SA, ESI mice were injected with Cre-GFP-AAV in the PrL and retrograde DIO-mCherry-AAV in the VTA (Fig. S4A, B). After virus expression, ESI mice were counterbalanced based on their behavioral performance (Fig. S4C–E) and assigned to receive vehicle or C21 injection. Then these mice were subjected to a heroin-seeking test 30 min after C21 i.p. injection. We found that ESI mice injected with vehicle or C21 did not differ in total active responses (Fig. S4F), total inactive responses (Fig. S4G), or total responses (Fig. S4H) during the heroin-seeking test. C21 injection also did not change the locomotor activity (Fig. S4I). These data demonstrate that C21 does not have an off-target effect.

Next, we took further steps to evaluate whether changes in heroin-seeking behavior are related to alterations of neuronal firing in PrL->VTA projecting neurons. GH and ESI mice that received double-viral injection (DIO-hM3Dq-mCherry-AAVrg into the VTA and Cre-GFP into the PrL) were injected with C21 (i.p.) after 14 days of abstinence, and

then euthanized for whole-cell patch-clamp recording of neuronal firing 30 min after C21 injection (Fig. 2A). Again, PrL->VTA projecting neurons were visualized under the microscope with both GFP and mCherry labeling (Fig. 2C), and the frequency of sAP was recorded. Consistently, we found that sAP frequency was lower in PrL->VTA projecting neurons from ESI mice that underwent heroin abstinence compared to the GH mice (Fig. 2H, ESI Veh vs GH Veh, two-way ANOVA followed by post hoc Tukey test, $p < 0.05$). C21 treatment counteracted the reduction of sAP frequency in PrL->VTA projection from ESI mice that underwent heroin abstinence (Fig. 2H, ESI C21 vs ESI Veh, $p < 0.05$). We also found that C21 treatment increased the sAP frequency in GH mice that underwent heroin abstinence (Fig. 2H, GH C21 vs GH Veh, $p < 0.05$). Moreover, C21 treatment did not induce any changes in sAP amplitude (Fig. S5A), half-width (Fig. S5B), rise time (Fig. S5C), and decay time (Fig. S5D) in PrL->VTA projection.

We then examined if chemogenetic manipulation-mediated effects on heroin seeking and sAP frequency also aligned with changes in excitatory synaptic functions. sEPSC frequency of PrL->VTA projecting neurons was significantly reduced in ESI Veh mice compared to GH Veh mice (Fig. 2I, ESI Veh vs GH Veh, two-way ANOVA followed by post hoc Tukey test, $p < 0.05$, Fig. S5E), replicating our initial results (Fig. 1F). C21 treatment increased the sEPSC frequency in both GH ($p < 0.05$) and ESI ($p < 0.05$) mice that underwent heroin abstinence. Intriguingly, C21 treatment increased sEPSC amplitudes in GH ($p < 0.05$) and ESI ($p < 0.05$) mice (Figs. 2J and S5F). Taken together, the hypoactive state of PrL->VTA projecting neurons may underlie ESI-potentiated heroin seeking, and this hypofunction may be driven by decreased excitatory synaptic transmission.

## Inhibition of PrL-VTA projection during abstinence potentiates heroin seeking in group-housed mice

To further explore the role of PrL->VTA projection in heroin seeking, we next used hM4D(Gi) DREADDs to inhibit the PrL->VTA projecting neurons. A mCherry-coupled and Cre-dependent hM4Di receptor-expressing AAV was bilaterally injected into the PrL of C57BL/6J GH mice. Retrograde Cre-AAVrg was injected into the VTA to initiate the expression of hM4Di receptors on the PrL->VTA projection. We validated this model by using the whole-cell patch-clamping method and by measuring c-Fos expression levels. We found that sAP frequency of PrL->VTA projecting neurons (labeled by mCherry, Fig. S6A) was notably inhibited after C21 bath application (10 μM[56], Fig. S6B). The inhibition of PrL->VTA projection was confirmed by fewer c-Fos positive cells after C21 injection (1 mg/kg, i.p.[57]) compared to the vehicle group (Fig. S6C, D). These data suggest that hM4D(Gi) selectively inhibited neuronal firing within the PrL->VTA projection.

Then, the role of PrL->VTA projection inhibition in heroin seeking was investigated. GH C57BL/6J mice received water SA training followed by jugular catheterization at the age of 8 weeks, and then were trained to self-administer heroin for 10 days (Figs. 3A and S7A, B).

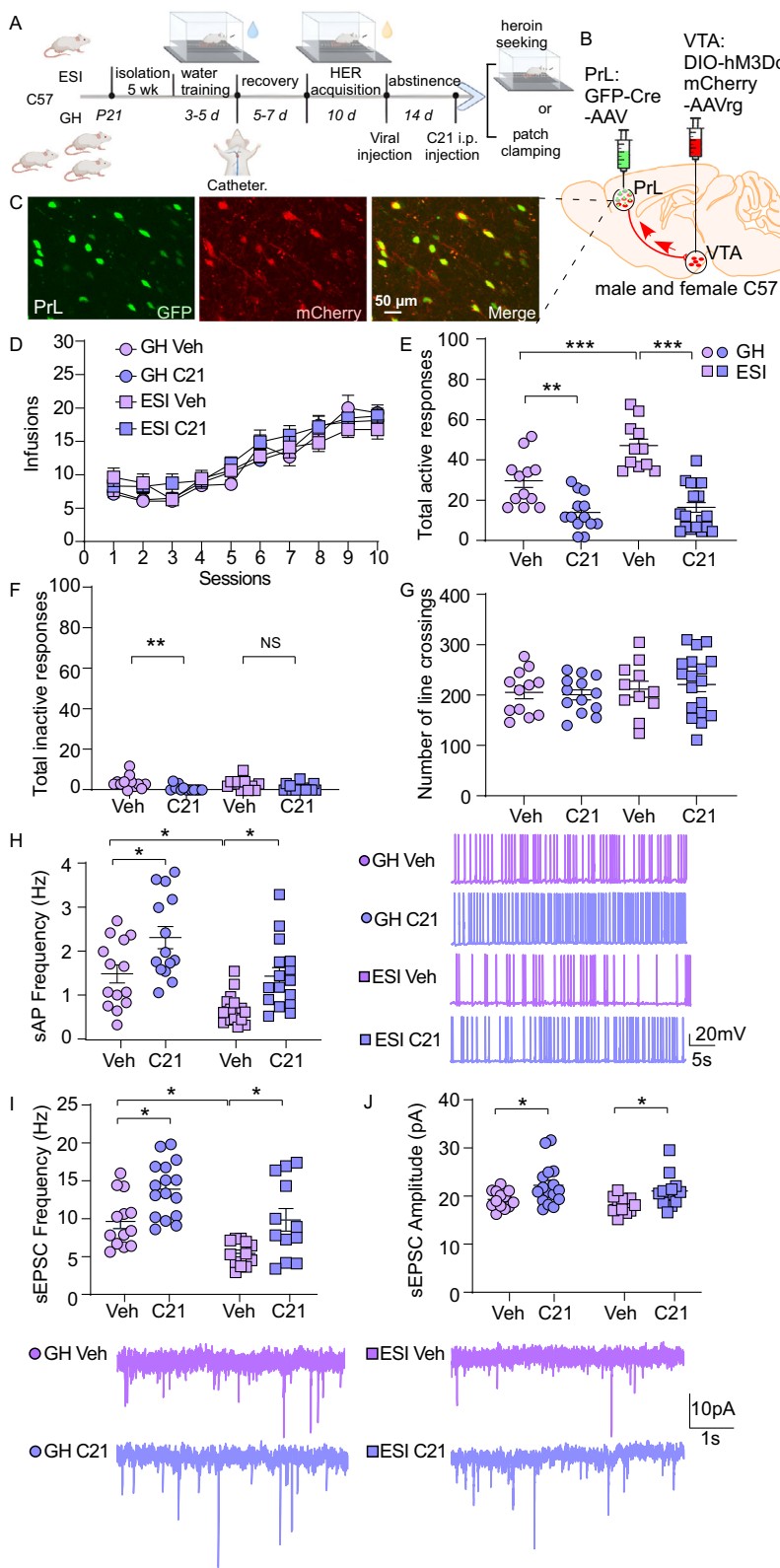

Then, mice were subjected to forced abstinence, during which time mice received double-viral injections (DIO-hM4Di-mCherry-AAV into the PrL and Cre-AAVrg into the VTA, Fig. 3B). After 14 days of abstinence and viral expression, mice were subjected to the heroin-seeking test. Mice were counterbalanced based on the averaged infusion numbers (Fig. 3C) and assigned to receive vehicle or C21 injections (1 mg/kg, i.p.[57]) 30 min before the heroin-seeking test. We found that the total active responses (Fig. 3D, $p < 0.001$, $t$-test) and total responses (Fig. S7C, $p < 0.01$) were significantly higher in C21-injected GH mice than in the vehicle group. Importantly, chemogenetic inhibition did not alter total inactive responses (Fig. 3E) and locomotor activity (Fig. 3F). There was no pre-existing bias in heroin SA (infusion numbers: Fig. 3C; total active and inactive responses: Fig. S7A, B) between C21 and vehicle groups.

**Fig. 2 | Chemogenetic activation of PrL->VTA projection attenuates ESI-potentiated heroin seeking. A** Experimental timeline for water training, heroin (HER) self-administration, virus injection, C21 intraperitoneally (i.p.) injection (1 mg/kg), behavioral testing, and electrophysiology in C57BL/6 mice. Schematic of Gq-DREADD injection site (**B**) and visualization of DREADD-expressing PrL->VTA projecting neurons in the PrL (**C**) of C57BL/6 mice. **D** Mean numbers of HER infusions per session during the acquisition of SA ($n = 15$ [GH Veh], 16 [GH C21], 14 [ESI Veh], and 21 [ESI C21] mice/group). GH Veh: group-housed vehicle; GH C21: group-housed C21; ESI Veh: early social isolated vehicle; ESI C21: early social isolated C21). Mean numbers of **E** total active responses ($n = 12$ [GH Veh], 13 [GH C21], 11 [ESI Veh], and 18 [ESI C21] mice/group, two-way ANOVA, $F_{1, 50 (stress)} = 12.99$, $p = 0.0007$, $F_{1, 50 (treatment)} = 69.67$, $p < 0.0001$, $F_{1, 50 (intercept)} = 7.24$, $p = 0.0097$) and **F** total inactive responses (Kruskal–Wallis test, $\chi^2(3) = 16.39$, $p < 0.001$) during heroin-seeking test.

**G** Mean number of line crossings during locomotion test ($n = 12$ [GH Veh], 13 [GH C21], 11 [ESI Veh], and 18 [ESI C21] mice/group. **H** Left: Graph showing the firing frequency of sAP in PrL->VTA projecting neurons across all groups ($n = 14$ [GH Veh], 14 [GH C21], 17 [ESI Veh], and 16 [ESI C21] cells from 3 mice/group, two-way ANOVA, $F_{1, 57 (stress)} = 21.33$, $p < 0.0001$, $F_{1, 57 (treatment)} = 18.26$, $p < 0.0001$). Right: representative sAP traces. Top: Bar graph showing the frequencies (**I**) and amplitudes (**J**) of sEPSCs in PrL->VTA projecting neurons ($n = 13$ [GH Veh], 16 [GH C21], 12 [ESI Veh], and 12 [ESI C21] cells from 3 mice/group, two-way ANOVA, I: $F_{1, 49 (stress)} = 16.49$, $p = 0.0002$, $F_{1, 49 (treatment)} = 18.14$, $p < 0.0001$; J: $F_{1, 49 (treatment)} = 11.14$, $p = 0.0016$). Bottom: Representative sEPSC traces. Data are expressed as mean ± SEM, two-way ANOVA, *$p < 0.05$, **$p < 0.01$, ***$p < 0.001$. **A** created in BioRender. Wang, Z. (2025) https://BioRender.com/6kfkeg5. Source data are provided as a Source data file.

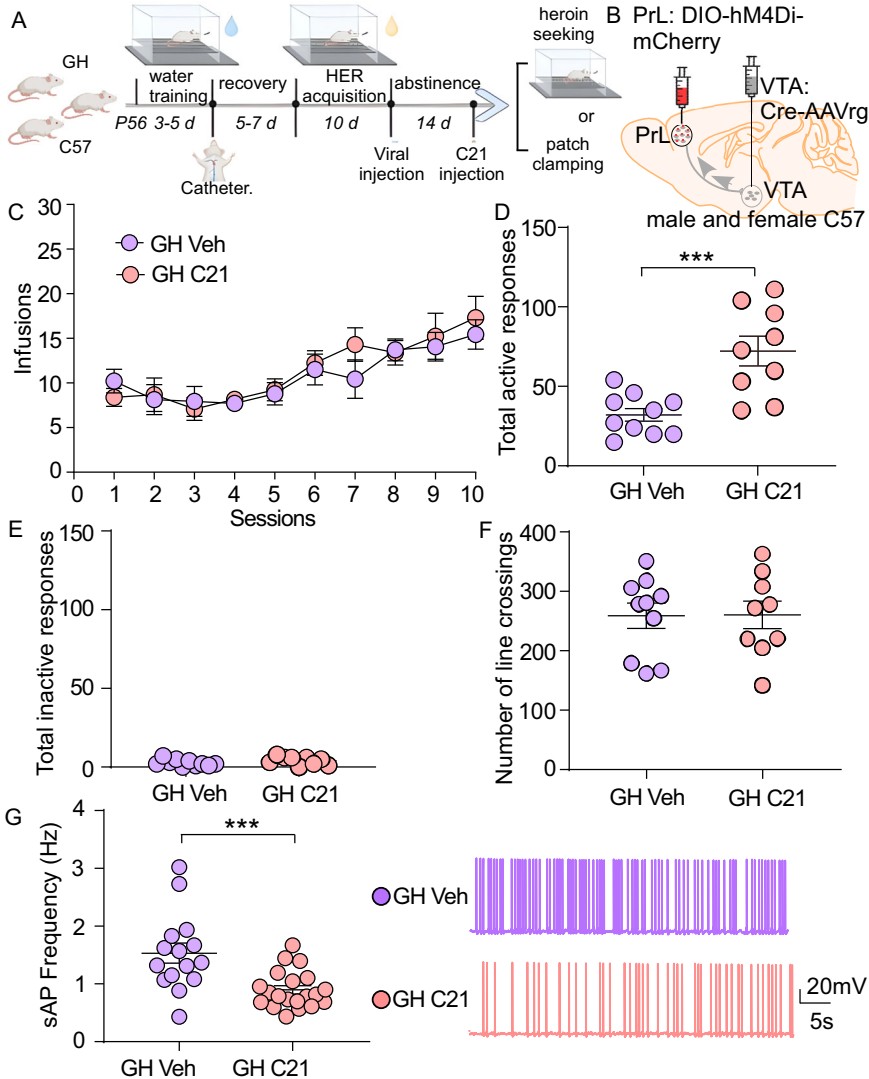

**Fig. 3 | Chemogenetic inhibition of PrL->VTA projection increases heroin seeking in GH mice. A** Experimental timeline for water training, heroin (HER) self-administration, virus injection, C21 i.p. injection (1 mg/kg), behavioral testing, and electrophysiology in GH C57BL/6 mice. **B** Schematic of Gi-DREADD injection site. **C** Mean numbers of heroin infusions per session during the acquisition of SA in GH mice ($n = 14$ [GH Veh] and 13 [GH C21] mice/group, GH Veh group-housed vehicle, GH C21 group-housed C21). Mean numbers of total active responses (**D**, $n = 10$ [GH Veh] and 9 [GH C21] mice/group, unpaired two-tailed $t$-test, $t_{17} = 4.08$, $p = 0.0008$)

and total inactive responses (**E**) during heroin seeking. **F** Mean number of line crossings during locomotion test ($n = 10$ [GH Veh] and 9 [GH C21] mice/group). **G** Left: Graph showing the firing frequency of sAP in PrL->VTA projecting neurons between GH Veh and GH C21 groups ($n = 15$ [GH Veh] and 20 [GH C21] cells from 4 mice/group, unpaired two-tailed $t$-test, $t_{33} = 3.72$, $p = 0.0007$). Right: representative sAP traces. Data are expressed as mean ± SEM, ***$p < 0.001$. **A** created in BioRender. Wang, Z. (2025) https://BioRender.com/6kfkeg5. Source data are provided as a Source data file.

Next, we evaluated whether changes in heroin-seeking behavior are related to alterations of neuronal firing in PrL->VTA projecting neurons. Some mice that received double-viral injections were treated with C21 (i.p.) after 14 days of abstinence and then euthanized (30 min after C21 injection) for whole-cell patch-clamp recording of neuronal firing (Fig. 3A). The PrL->VTA projecting neurons were visualized under the microscope with mCherry labeling, and the frequency of sAP was recorded. We found that C21 treatment significantly inhibited the sAP frequency in PrL->VTA projecting neurons after 14 days of heroin abstinence in GH mice (Fig. 3G, $p < 0.001$). Collectively, these data suggest that inhibition of the PrL->VTA projection can potentiate heroin seeking in GH C57BL/6J mice.

## ESI stress and heroin abstinence alter transcriptional profile in PrL->VTA projecting neurons: convergent effect

The bidirectional interactions between neuronal activity and gene expression are important for shaping behaviors[58,59]. Therefore, we next investigated the transcriptional changes underlying the hypoactive state of the PrL->VTA projection induced by heroin abstinence and ESI stress interactions. RNA enriched in the PrL->VTA projection was isolated using the retrograde-TRAP method in GFP-L10a mice, in which the expression of the eGFP::L10a fusion gene is blocked by a loxP-flanked STOP cassette (three copies of the SV40 polyA signal)[60] (Fig. S8A). When combined with Cre-expressing viruses, successful Cre-mediated excision leads to constitutive expression of the eGFP-tagged form of the L10a ribosomal subunit in Cre-expressing cells (Fig. S8A). To validate this model, the AAVrg-Cre virus was injected into the VTA of GFP-L10a mice (Fig. S8B). After viral expression, brain slices containing PrL were collected and stained with the pyramidal neuron marker CaMKIIa[31,61]. We found that GFP-positive neurons, representing PrL neurons projecting to the VTA, colocalized with CaMKIIa (Fig. S8C). This is consistent with the findings that pyramidal neurons constitute the dominant cell type in the PrL->VTA circuits[49,62,63]. Moreover, brain tissues containing PrL were incubated with a GFP-specific antibody to selectively pull down mRNA from GFP-L10a-expressing cells (Fig. S8D). qPCR results confirmed that compared to total cell populations (i.e., input), GFP-L10a-expressing cells showed a much higher level of *Pou3f1* gene, which is enriched in the PrL->VTA projection (Fig. S8E)[37,38], as well as a much lower level of *Tnnc1* gene, which is enriched in PrL projections to other regions (e.g., NAc and amygdala)[37,38] (Fig. S8E). Furthermore, we performed RNA-seq to identify the genome-wide transcriptional differences by comparing RNA enriched in PrL->VTA projection (all TRAP-seq data from this study) to total RNA (bulk RNA-seq data from mouse PrL). Consistent with previous publication[38], we found that PrL->VTA projection has a distinct transcriptional profile compared to total bulk RNA, with 1374 downregulated and 3525 up-regulated genes (Supplementary Data 2) using the cutoff of adjusted $p$ value (FDR) $< 0.05$ and $|Log2FC| > 1$; or 1283 downregulated and 3293 up-regulated genes when using a more stringent criterion FDR $< 0.01$ and $|Log2FC| > 1$. A heatmap generated with differentially expressed gene (DEGs) with FDR $< 0.05$ showed that PrL->VTA projection-enriched RNA samples clustered together and were separated from total RNA samples (Fig. S8F). We further compared the PrL->VTA projection-enriched genes list (with FDR $< 0.05$) in our current study to the gene list from Murugan et al.[38], and found significant overlap between these two gene lists (Fig. S8G, H). These data suggest that the TRAP method can reliably capture translating RNAs enriched in the PrL->VTA projection.

Using this validated TRAP technique, we further explored the genome-wide gene expression changes in PrL->VTA projecting neurons induced by ESI stress, heroin abstinence, and their interaction. GH or ESI GFP-L10a mice were subjected to saline or heroin SA (Fig. S9A–E). AAVrg-Cre virus was injected into the VTA during abstinence (Fig. S9B). After viral expression, mice were euthanized for TRAP followed by RNA-seq study (Fig. S9A). We first investigated the DEGs

dependent on each factor (stress factor [GH ESI] or drug factor [SAL HER]). We identified that 195 DEGs were impacted by ESI stress (DEG$^{ESI}$: ESI vs GH, Fig. 4A, Supplementary Data 3), with 121 downregulated transcripts and 74 up-regulated transcripts. Heatmap showed that ESI samples were differentiated from GH samples by unsupervised clustering based on these 195 DEG$^{ESI}$ (Fig. S10A). We also found that 374 DEGs were impacted by abstinence from heroin SA (DEG$^{HER}$: HER vs SAL, Fig. 4B, Supplementary Data 4), with 120 downregulated and 254 up-regulated genes. The heatmap showed that HER samples were clustered together and away from SAL samples (Fig. S10B). Most transcripts differentially regulated by ESI stress or heroin fell into similar protein categories, such as enzymes and enzyme regulators, genetic information processing proteins, scaffold/adapter proteins, and cytoskeletal proteins (Fig. S10C).

Because protein categories were largely comparable, we further explored whether ESI stress- and heroin-induced DEGs have similar biological functions. To functionally categorize these DEGs, we performed Gene Ontology (GO) analysis with a focus on Biological Process. We found that ESI stress-induced DEGs could be assigned to some unique biological processes, such as regulation of histone methylation and chromatin modification (Fig. 4C), suggesting that epigenetic regulation of gene expression was altered by ESI stress. This is consistent with studies showing that early life adversity induces epigenetic changes[34,64,65]. Some other unique biological processes were neurotransmitters, synaptic vesicles, and cellular macromolecule localization, which regulate synaptic transmission and neuronal activity (Fig. 4C), in agreement with our data showing ESI stress disrupted neuronal activity and excitatory synaptic transmission (Fig. 1D, F).

Additionally, we found that DEGs in PrL->VTA projection induced by long-term abstinence from heroin SA were enriched in biological processes that are known to have critical roles in drug addiction (e.g., response to growth factor[66], MAPK cascade[67,68], TGF-β receptor signaling pathway[69,70]) and cytoskeleton organization (e.g., cell adhesion, actin cytoskeleton organization, and axon extension) (Fig. 4D). Furthermore, synaptic enrichment analyses identified 41 DEG$^{HER}$ genes as synaptic genes (while DEG$^{ESI}$ showed insignificant synapse enrichment), with the postsynaptic ribosome and presynaptic ribosome as the two significantly enriched subcellular components (Fig. 4E). These data confirm our original findings that heroin abstinence altered neuronal activity and excitatory synaptic transmission (Fig. 1D, F) and echo previous publications reporting opioid-induced morphological changes in cortical pyramidal neurons[71,72].

Interestingly, we found several shared biological processes between DEG$^{ESI}$ and DEG$^{HER}$ that regulate neuronal morphology (e.g., anatomical structure morphogenesis and small GTPase-mediated signal transduction), cell communication and synaptic transmission (e.g., regulation of transport and cation transport), as well as oxidative stress (Fig. 4C, D), which are all critical in controlling synaptic and neuronal function. However, most of the genes within the shared BP categories were non-overlapping (Supplementary Data 5). These data suggest that ESI stress and heroin abstinence may convergently alter the expression of genes within similar BP categories instead of altering the same genes to potentiate the hypoactive state of PrL->VTA projection.

To explore the transcriptional overlap between ESI stress- and heroin abstinence-induced DEGs, we first used rank–rank hypergeometric overlap (RRHO) analysis to compare threshold-free gene expression patterns. We found some overlap between downregulated genes (Fig. 4F, G, hot spots in upper right quadrants). However, ESI stress and heroin abstinence induced unique gene expression patterns, such as many genes up-regulated by ESI stress were downregulated by heroin abstinence (Fig. 4F, G, hot spots in upper left quadrants), and some genes downregulated by ESI stress were up-regulated by heroin abstinence (Fig. 4F, G, hot spots in lower right quadrants). We then compared the two DEGs lists and found 11 overlapping genes (Figs. 4H and S10D); most overlapping genes were downregulated by both ESI stress and heroin

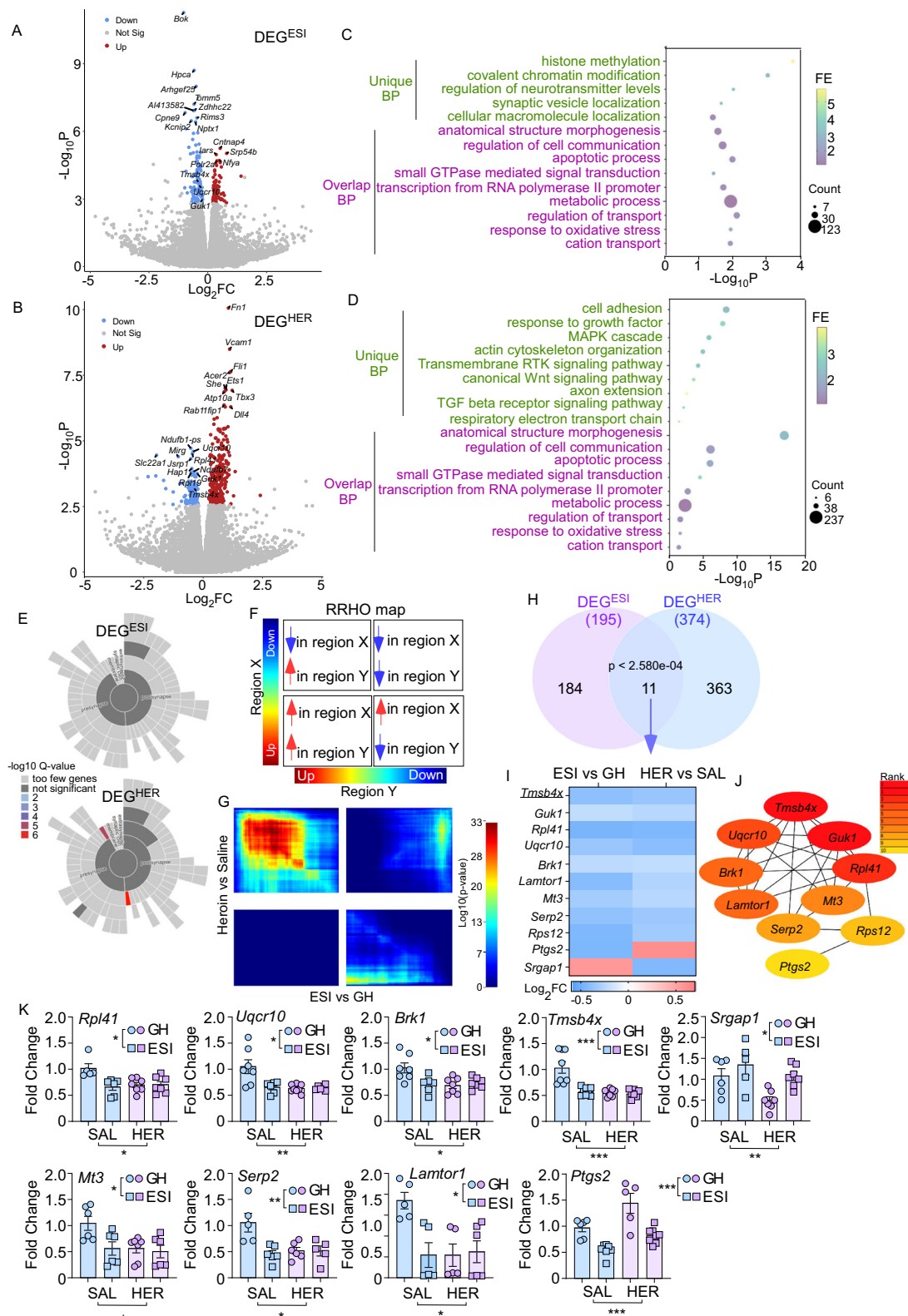

abstinence (Fig. 4I). We also validated the change of the 11 overlapping genes using qPCR (Fig. 4K; SA behavior data for mice used in qPCR were shown in Fig. S9). Furthermore, Cytohubba gene network analysis showed that *Tmsb4x* is one of the top-ranked hub genes within this network (Fig. 4J, rank = 1), and the reduction of *Tmsb4x* is the same in both male and female mice (Fig. S11A).

We next validated the role of *Tmsb4x, encoding* thymosin β4 (Tβ4), in ESI stress-potentiated PrL->VTA projection hypofunction and

heroin seeking. *Tmsb4x* is enriched in the PFC[73], and extracellular Tβ4 can be internalized into cells to trigger intracellular responses[74]. Therefore, we microinjected mouse recombinant Tβ4 in the PrL of ESI mice during the last 7 days of abstinence following heroin SA (Figs. 5A and S12A–C). We found that Tβ4 infusions (20 μg/mL, 1 μL/hemisphere[75]) into the PrL during abstinence attenuated the total active responses (Fig. 5C) and total responses (Fig. S12D) during the heroin-seeking test in ESI mice with a history of heroin SA followed by

**Fig. 4 | Heroin and ESI stress have convergent effects on PrL->VTA projection transcription.** Volcano plot showing differentially expressed genes (DEGs) induced by ESI (**A**, DEG^ESI: ESI vs GH) or heroin (HER) abstinence (**B**, DEG^HER: HER vs SAL) in PrL->VTA projection. Gene Ontology (GO) pathway enrichment analysis for DEGs induced by ESI stress (**C**, ESI vs GH) or HER abstinence (**D**, HER vs SAL). **E** Sunburst plot representing cellular component enrichment analysis of ESI- or HER-induced DEGs. Higher red intensities are associated with more significant enrichments. **F**, **G** Rank–rank hypergeometric overlap (RRHO) plot indicating the degree of overlap, or transcriptional concordance, between the ESI stress and HER abstinence effect on gene transcription in PrL->VTA projection. **H** Venn diagram showing the overlap between DEGs caused by ESI stress (DEG^ESI: ESI vs GH) and HER abstinence (DEG^HER: HER vs SAL). *P* value is calculated from the hypergeometric probability formula. **I** Heatmap of Log₂FC for the overlapping genes between DEG^ESI (ESI vs GH) and DEG^HER (HER vs SAL). **J** Cytohubba gene network analysis of overlapping DEGs caused by ESI stress and HER abstinence. **K** qPCR results for the DEG^ESI and DEG^HER overlapping genes (*n* = 5–8 mice/group: *Rpl41*: 5 [GH SAL], 6 [ESI SAL], 7 [GH HER] and 7 [ESI HER]; *Uqcr10*: 7 [GH SAL], 6 [ESI SAL], 8 [GH HER] and 6 [ESI HER]; *Brk1*: 7 [GH SAL], 6 [ESI SAL], 7 [GH HER] and 7 [ESI HER]; *Tmsb4x*: 8 [GH SAL], 8 [ESI SAL], 8 [GH HER] and 8 [ESI HER]; *Srgap1*: 6 [GH SAL], 5 [ESI SAL], 7 [GH HER] and 7 [ESI HER]; *Mt3*: 6 [GH SAL], 6 [ESI SAL], 7 [GH HER] and 6 [ESI HER]; *Serp2*: 5 [GH SAL], 6 [ESI SAL], 6 [GH HER] and 5 [ESI HER]; *Lamtor1*: 5 [GH SAL], 5 [ESI SAL], 5 [GH HER] and 6 [ESI HER]; *Ptgs2*: 6 [GH SAL], 7 [ESI SAL], 5 [GH HER] and 8 [ESI HER]). Data are expressed as mean ± SEM, two-way ANOVA, *p < 0.05, **p < 0.01, ***p < 0.001. Source data are provided as a Source data file.

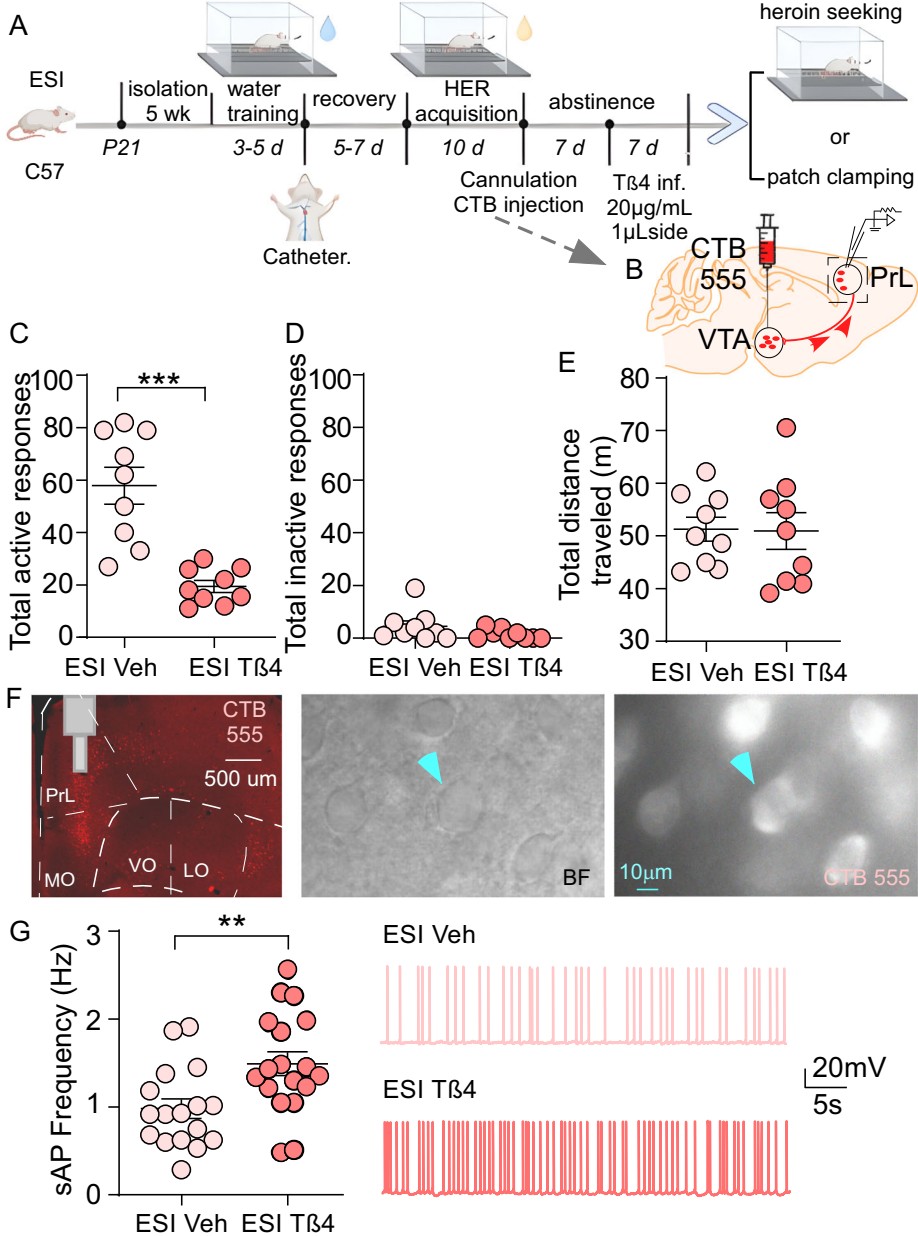

**Fig. 5 | Intra-PrL infusion of thymosin β4 (Tβ4) attenuates heroin-seeking behavior in ESI mice. A** Experimental timeline for water training, heroin (HER) self-administration, cannulation, Tβ4 intra-PrL infusion (20 μg/mL, 1 μL /hemisphere), behavioral testing, and electrophysiology in ESI C57BL/6 mice. **B** Schematic of the CTB 555 injection site in the VTA. Mean numbers of total active responses (**C**, n = 9 mice/group, unpaired two-tailed *t*-test, $t_{16}$ = 5.2, *p* < 0.0001) and total inactive responses (**D**) during heroin seeking. **E** Mean numbers of distance traveled (m) during locomotion test (*n* = 9 mice/group). **F** Visualization of PrL->VTA projection in the PrL during patch-clamp recording. MO, VO, LO: orbital cortex (medial, ventral, and lateral). **G** Left: Graph showing firing frequency of sAP in PrL->VTA projection between ESI Veh and ESI Tβ4 groups (*n* = 17 [ESI Veh]–18 [ESI Tβ4] cells from 3 mice/group, unpaired two-tailed *t*-test, $t_{33}$ = 2.91, *p* = 0.0065). Right: representative sAP traces. Data are expressed as mean ± SEM, **p < 0.01, ***p < 0.001. **A** created in BioRender. Wang, Z. (2025) https://BioRender.com/6kfkeg5. Source data are provided as a Source data file.

abstinence, without affecting the inactive responses and locomotion activity (Fig. 5D, E). There was also no pre-existing bias in heroin intake between Tβ4 and vehicle groups (Fig. S12A–C). Using CTB retrograde labeling (Fig. 5B), we also found that Tβ4 treatment recovered the decrease of sAP frequency in PrL->VTA projecting neurons in ESI mice that underwent heroin abstinence (Fig. 5F, G). These data suggest that ESI stress and heroin abstinence may convergently disrupt PrL->VTA projection neuronal function through Tβ4-involved mechanisms, which contribute to ESI-potentiated heroin seeking.

### ESI stress and heroin abstinence alter transcriptional profile in PrL->VTA projecting neurons: interaction effect

In addition to the convergent effect of ESI stress and heroin abstinence on gene expression, the interaction effect of the two factors (i.e., the effects of the two factors that are not additive[76]) could also contribute to ESI-induced relapse vulnerability. To explore the genes that were specifically affected by the interaction between ESI stress and heroin, we used a likelihood ratio test (LRT) analysis by comparing the full model (stress + drug + stress:drug) to the reduced model (stress + drug). ESI stress and heroin interaction induced 214 DEGs (Fig. 6A, Supplementary Data 6, hereafter referred to as "interaction DEGs" [DEG$^{inter}$]). Most interaction DEGs fall into protein categories such as enzymes and enzyme regulators, genetic information processing proteins, cytoskeletal proteins, cell adhesion and cell junction proteins, and intercellular signal molecules (Fig. S10C).

Functional annotation analysis revealed that interaction DEGs were enriched in biological process categories (e.g., metabolic process, biogenesis, cell cycle, and DNA damage/repair processes) and pathways (e.g., deubiquitination, RNA/DNA metabolism and synthesis, DNA damage responses, and cell cycle) that are closely related to DNA synthesis, DNA damage and repair processes (Fig. 6B). To identify the hub genes, we integrated the top 40 interaction DEGs with network enrichment analysis (Fig. 6C, D). We found that these top-ranking genes were enriched in the regulation of cell cycle, such as spindle process, microtubule organization, MCM complex, and DNA replication (Fig. 6C). Network analysis identified three major clusters (Fig. 6D). One cluster included 3 transcription factors, among which *FosB*[77] and *Nr4a2*[78,79] are known to play key roles in drug addiction. One cluster included two genes encoding the enzyme beta-secretase 2 (*Bace2*) and Hexokinase 2 (*Hk2*). The biggest cluster consisted of 10 genes regulating cell cycle, microtubule cytoskeleton, and G1/S cell cycle control, further supporting the functional annotation results. Among these, *Mcm3* and *Mcm7* were recognized as hub genes (interaction score = 0.999, identified by String, Fig. 6D, E, Supplementary Data 7). qPCR validation results further showed that there was a significant stress X drug interaction effect on the mRNA level of *Mcm3* and *Mcm7* in the PrL->VTA projection (Fig. 6F). This significant interaction effect is true in both male and female mice (Fig. S11B, C). Post hoc analysis revealed that *Mcm3* ($p < 0.01$) and *Mcm7* ($p < 0.001$) levels were significantly increased in the ESI HER group compared to the GH HER group, and this increase was mainly driven by the reduced expression in the GH HER group (Figs. 6F and S11B, C). Similarly, after 14 days of abstinence followed by SA (Fig. S13A–E), the protein levels of Mcm3 ($p < 0.0001$) and Mcm7 ($p < 0.01$) within the PrL->VTA projection (labeled by CTB 555, Fig. S13B) were significantly affected by the stress × drug interaction (Fig. 6G–I), with an obviously higher expression level in the ESI HER group compared to the GH HER group (Fig. 6I).

Mcm3 and Mcm7 are part of the minichromosome maintenance (MCM) protein complex, which governs cell cycles by gating DNA replication licensing[80,81] and also regulates DNA damage and repair[82]. However, in post-mitotic neurons, re-entering the cell cycle leads to neuronal dysfunction (e.g., apoptosis[83,84]). It is possible that, in GH mice, reduced *Mcm3* and *Mcm7* expression levels in PrL->VTA projecting neurons diminish MCM complex function to restrict DNA replication and cell cycle re-entry, which may minimize heroin-induced

neuronal dysfunction. This reduction of *Mcm3* and *Mcm7* gene expression in response to heroin abstinence was impaired by prior exposure to ESI stress. To test this hypothesis, we constructed an AAV vector carrying sgRNA targeting *Mcm3* and *Mcm7* genes to achieve CRISPR-Cas9-mediated conditional knockdown (cKD) of *Mcm3* and *Mcm7*. Retrograde AAVrg-Cre was injected into the VTA of Cas9$^{flox/flox}$ mice to drive the Cre-dependent GFP-fused Cas9 protein expression in the PrL->VTA projection. In the meantime, AAV-spsg*Mcm3*-spsg*Mcm7*-mcherry was delivered in the PrL to mediate the knockdown of *Mcm3* and *Mcm7* gene expression (Fig. 7A). The possible effects of CRISPR-mediated off-site mutagenesis were excluded (Fig. S14). Fluorescence in situ hybridization indicated that *Mcm3* (Fig. 7B, C) and *Mcm7* (Fig. 7D, E) mRNA levels in PrL->VTA projection infected with spsgRNA (mCherry$^+$GFP$^+$) were significantly decreased, as compared with the control group (scramble), demonstrating specific knockdown of *Mcm3* and *Mcm7* mRNA levels in the PrL->VTA projection.

Using these validated tools, we performed the same viral surgery mediating *Mcm3* and *Mcm7* cKD in the PrL->VTA projecting neurons during abstinence following heroin SA in ESI mice (Figs. 7F and S15A–C). We found that *Mcm3* and *Mcm7* cKD attenuated total active responses (Fig. 7G) and total responses (Fig. S15D) during the heroin-seeking test, without changing inactive responses or locomotion activity (Fig. 7H, I). Pre-existing heroin intake bias between sgRNA and scramble groups was also absent (Fig. S15A–C). Another cohort of ESI mice receiving the same viral surgery during abstinence after heroin SA was directly euthanized for the recording of spontaneous AP in the mCherry$^+$GFP$^+$ neurons in the PrL, without a drug-seeking test. We found that *Mcm3* and *Mcm7* cKD increased the sAP frequency in PrL->VTA projecting neurons (Fig. 7J). These data suggest that *Mcm3* and *Mcm7* play a critical role in regulating ESI-potentiated drug-seeking behavior via regulating neuronal function in the PrL->VTA circuit. All together, these findings suggest that targeting *Tmsb4x* and the cell cycle-regulating MCM complex in the PrL->VTA circuit could represent an approach to modulating ESI-induced vulnerability to opioid relapse (Fig. 7K).

## Discussion

### ESI stress and heroin abstinence induced neuronal dysfunction in the prefrontal cortex

The functional development of neuronal networks is primarily determined by environmental stimuli after their formation[85,86]. Adolescence is a critical period during which neuronal circuits are reformed by experience[6-9]. Therefore, social isolation during adolescence has a profound impact on the maturation of neuronal circuits[17,87], contributing to the increased risk for psychiatric disorders[15,18]. We found that ESI stress disrupts neuronal function in the PrL to VTA circuit by reducing the frequency of sAP and excitatory synaptic transmission (Fig. 1D, F) in the PrL->VTA projecting neurons. Similarly, studies have shown that social isolation stress after weaning alters neuronal function in the PFC pyramidal neurons, such as reduced neuronal activity[35], deterioration in AP properties and reduction in excitatory synaptic inputs[32], reduced density of dendritic spines[88], as well as abnormal cortical processing in response to VTA cell stimulation[89]. These studies, together with our data, suggest that ESI stress can compromise pyramidal neuron function in the PFC to create vulnerable states, which may potentially contribute to the susceptibility to psychiatric traits in the future.

Interestingly, we found that prolonged abstinence from heroin SA also inhibited firing rate and reduced excitatory synaptic transmission in PrL->VTA projecting neurons (Fig. 1D, F). Studies have shown that opioids alter the excitatory synaptic transmission in PFC pyramidal neurons. For example, re-exposure to heroin-associated cues reduces the AMPAR/NMDAR ratio[27]; prolonged abstinence from remifentanil SA causes hypoactive states in dorsomedial PFC neurons, which was attributed to reduced AMPAR-mediated excitatory synaptic transmission in female mice[31]; protracted abstinence from fentanyl SA decreases AMPAR- and NMDAR-mediated postsynaptic transmission[90].

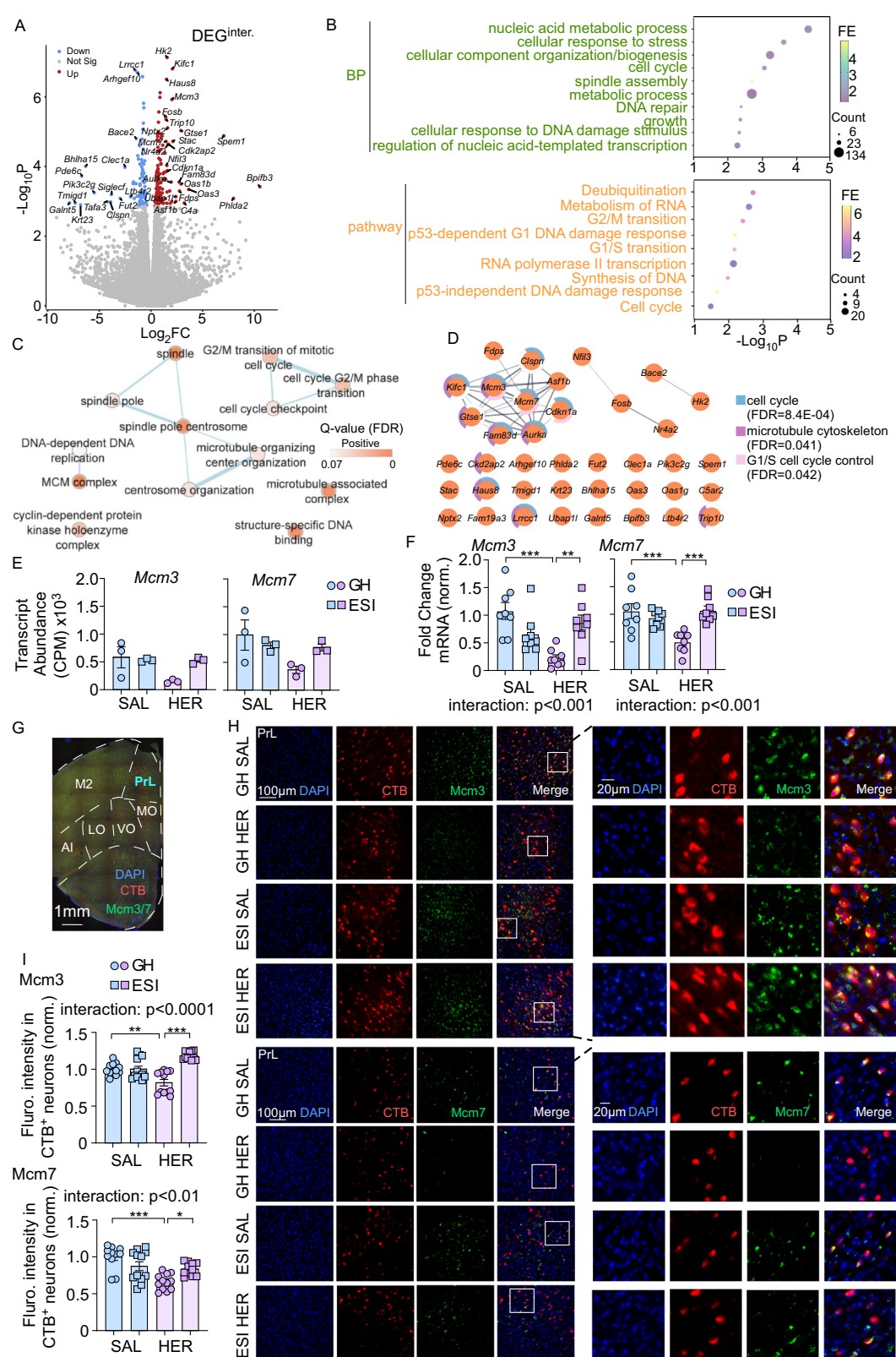

It is likely that prior exposure to ESI stress aggravates the heroin-induced neuronal dysfunction in the PrL->VTA projection (Fig. 1D–F) to potentiate heroin-seeking behavior (Fig. 2E)[21]. In addition, although the aggravated hypoactivity of PrL->VTA projecting neurons is due to reduced sEPSC frequencies (indicating a possible presynaptic effect is involved (Fig. 1F–H)) and slightly reduced neuronal excitability (Fig. 1E), we cannot rule out the role of inhibitory synaptic transmission, especially when GABAergic neurons in PrL are involved in opioid reward formation[91].

## Chemogenetic manipulation of PrL->VTA projection during abstinence regulates ESI-potentiated heroin seeking

Many studies have shown that manipulation of neuronal activity in different PFC subregions affects opioid-seeking behavior. GABA

**Fig. 6 | Heroin abstinence and ESI stress have interaction effects on PrL->VTA projection transcription. A** Volcano plot showing the DEGs induced by ESI stress and heroin (HER) abstinence interaction (DEG$^{inter}$) in PrL->VTA projections (FC $\approx \frac{ESI-HER}{ESI-SAL} = \frac{ESI-HER}{GH-HER}$). **B** Gene Ontology (GO) pathway enrichment analysis for DEGs induced by ESI stress and heroin abstinence interaction. **C, D** Network enrichment analysis of the top 40 DEG$^{inter}$. **E** Transcript quantification of *Mcm3* and *Mcm7* genes in RNA-Seq (counts per million, CPM) in PrL->VTA projection across all groups (*n* = 3 mice/group). **F** qPCR validation of changes in *Mcm3* and *Mcm7* genes expression across all groups (*n* = 8 mice/group; *Mcm3*: two-way ANOVA, $F_{1, 28 \text{ (interaction)}}$ = 18.41, *p* = 0.0002; *Mcm7*: two-way ANOVA, $F_{1, 28 \text{ (interaction)}}$ = 15.63, *p* = 0.0005). **G, H** Representative images of Mcm3 and Mcm7 staining in brain sections containing PrL (PrL: prelimbic cortex; M2: secondary motor cortex; MO, VO, LO: orbital cortex [medial, ventral and lateral]; AI: anterior insula) showing co-localization of CTB 555 (injected in VTA, red), Mcm3 (Green, **H** upper panel) or Mcm7 (green, **H** bottom panel) and DAPI (blue) in C57BL/6 mice. **I** Quantification of Mcm3 (*n* = 12 [GH SAL], 12 [ESI SAL], 12 [GH HER] and 11 [ESI HER] images from 4 mice/group, two-way ANOVA, $F_{1, 43 \text{ (interaction)}}$ = 28.34, *p* < 0.0001) and Mcm7 (*n* = 11 [GH SAL], 12 [ESI SAL], 12 [GH HER] and 11 [ESI HER] images from 4 mice/group, two-way ANOVA, $F_{1, 42 \text{ (interaction)}}$ = 11.58, *p* < 0.01) immunofluorescence intensity in PrL->VTA projection (CTB+). Data are expressed as mean ± SEM, two-way ANOVA, **p* < 0.05, ***p* < 0.01, ****p* < 0.001. Source data are provided as a Source data file.

receptor agonist infusion into the PrL enhances cue-induced reinstatement of heroin seeking[36]. However, others suggested that inactivation of the PrL or IL attenuates cue-, heroin-, or context-induced heroin reinstatement[23–25]. These discrepancies can partially be attributed to distinct diffusive projections, especially when different projections from the PrL play distinct roles in drug seeking. For example, PrL projections to the NAc precipitate cocaine and cue-induced reinstatement[92], whereas PrL projections to the rostromedial tegmental nucleus (RMTg) suppress cue-induced reinstatement of cocaine seeking[49]. Here, we showed that activation of PrL projections to the VTA attenuates heroin seeking in both ESI and GH mice (Fig. 2E), and inhibition of this projection in GH mice recapitulates the augmented heroin-seeking behavior by ESI stress (Fig. 3D), suggesting that PrL->VTA projection hypofunction plays a causal role in ESI-potentiated heroin seeking. In fact, the PrL->VTA projection has been implicated in regulating other motivational behaviors and responding to stress. For example, social isolation stress during adolescence causes PrL->VTA projection hypofunction to reduce social interaction in female mice[35] (sex differences are discussed below).

It is possible that PrL->VTA projection hypoactivity affects glutamate release in the VTA, which is critical for heroin seeking[93]. Three major excitatory inputs into VTA DA neurons include the mPFC, the pedunculopontine region, and the subthalamic nucleus. These excitatory inputs control the firing pattern of DA neurons, switching from a pacemaker-like manner into bursting mode[41,42]. Studies have shown that morphine withdrawal caused a presynaptic inhibition of glutamatergic EPSCs in VTA DA neurons[94]. Echoing these findings, our data indicate that PrL->VTA projection hypofunction may contribute to the inhibition of glutamatergic EPSCs in VTA DA neurons at the baseline level. However, it is also possible that the PrL->VTA circuit sends projections to VTA inhibitory neurons[40], thereby inhibiting VTA DA neuronal activity. In this case, the PrL->VTA projection hypofunction may reduce VTA inhibitory neuronal activity, which disinhibits VTA DA neurons during heroin seeking. Future studies are needed to elucidate the cell type-specific control of prefrontal innervation in the VTA at the baseline level and during drug-seeking tests.

Chemogenetic activation of PrL->VTA projection is accompanied by increased excitatory input in the PrL->VTA projecting neurons, in particular the increased sEPSC frequency (Fig. 2I), indicating a potential presynaptic effect. Consistently, studies showed that DREADD activation results in increased sEPSC amplitude and/or frequency[31,51,95]. These studies suggest that hM3Dq activation can have pre- and/or postsynaptic effects. hM3Dq activation increases the intracellular Ca$^{2+}$ and protein kinase C levels to induce neuronal activation, which can consequently activate CaMKII. Interestingly, activated CaMKII is not only a central molecular organizer of postsynaptic plasticity, but also remodels the presynaptic inputs by altering quantal content[96]. Moreover, DREADD-induced neuronal activity changes may alter the function of local tripartite synapses[97], which include astrocytes, the endogenous regulators of synaptic transmission[98,99], including presynaptic vesicle release[100]. These potential mechanisms may underlie the enhanced excitatory synaptic transmission after hM3Dq activation, which recovers the hypoactivity of the PrL->VTA projection after heroin abstinence and attenuates heroin seeking in both GH and ESI mice. We noticed that PrL->VTA activation also attenuated inactive responses during the drug-seeking test in GH but not ESI mice (Fig. 2F), while locomotion activity was unaltered. Additional research is necessary to elucidate whether activation of the PrL->VTA circuit has a universal effect in attenuating operant behavioral responding.

## ESI stress and heroin abstinence induce transcriptional changes in PrL->VTA projection

**ESI stress and heroin abstinence convergently reduce Tmsb4x gene expression.** Studies revealed that early life stress can directly alter transcriptional profiles across reward circuitries, including the PFC[34]. What are the molecular mechanisms underlying the ESI-potentiated hypofunction of PrL->VTA projecting neurons after heroin abstinence? Using the TRAP-seq method, our studies showed that ESI stress and heroin abstinence can both affect the expression of genes related to structural morphology, cell communication, and synaptic transmission. *Tmsb4x* is one of the key hub genes that is downregulated by both ESI stress and heroin abstinence (Fig. 4K). *Tmsb4x* is enriched in the brain, including the PFC[73,101]. In addition, as a moonlighting protein, thymosin β4 (encoded by *Tmsb4x*) is involved in the regulation of F/G-actin dynamics, tissue regeneration, cell migration, anti-inflammation, and many other cellular functions[102,103]. Among these physiological functions, the regulation of actin dynamics is probably one of the most important effects. As a G-actin sequestering peptide, thymosin β4 maintains a globular actin pool for the remodeling of neuronal processes and function[101,104]. This regulation of actin dynamics may underlie the convergent effect of ESI stress and heroin abstinence on impairing the excitatory synaptic transmission and neuronal activity in PrL->VTA projecting neurons. The attenuation of heroin-seeking behavior after thymosin β4 infusion into the PrL of ESI mice further supports that thymosin β4-mediated synaptic remodeling may directly contribute to ESI-potentiated heroin seeking. In fact, thymosins have been reported to directly modulate excitatory synaptic transmission[105]. One caveat is that thymosin β4 infusion in PrL can affect its function beyond the PrL->VTA projecting neurons. Another limitation is that thymosin β4 impacts many cellular processes. Under our experimental setting, it is hard to conclude whether the thymosin β4 effect is selective to ESI-potentiated heroin seeking or could be generalized to other drugs. Further studies need to elucidate how thymosin β4 in other cells (neurons or glia) within the PrL may affect ESI-potentiated heroin seeking and beyond.

We also found that ESI stress and heroin abstinence can convergently affect the expression of genes related to metabolic processes and oxidative stress (Fig. 4C, D). This is in line with studies showing that chronic social isolation stress affects oxidative stress, and that oxidative stress is one of the key mediators for the social isolation stress-induced phenotypes related to neuropsychiatric disorders[106–108].

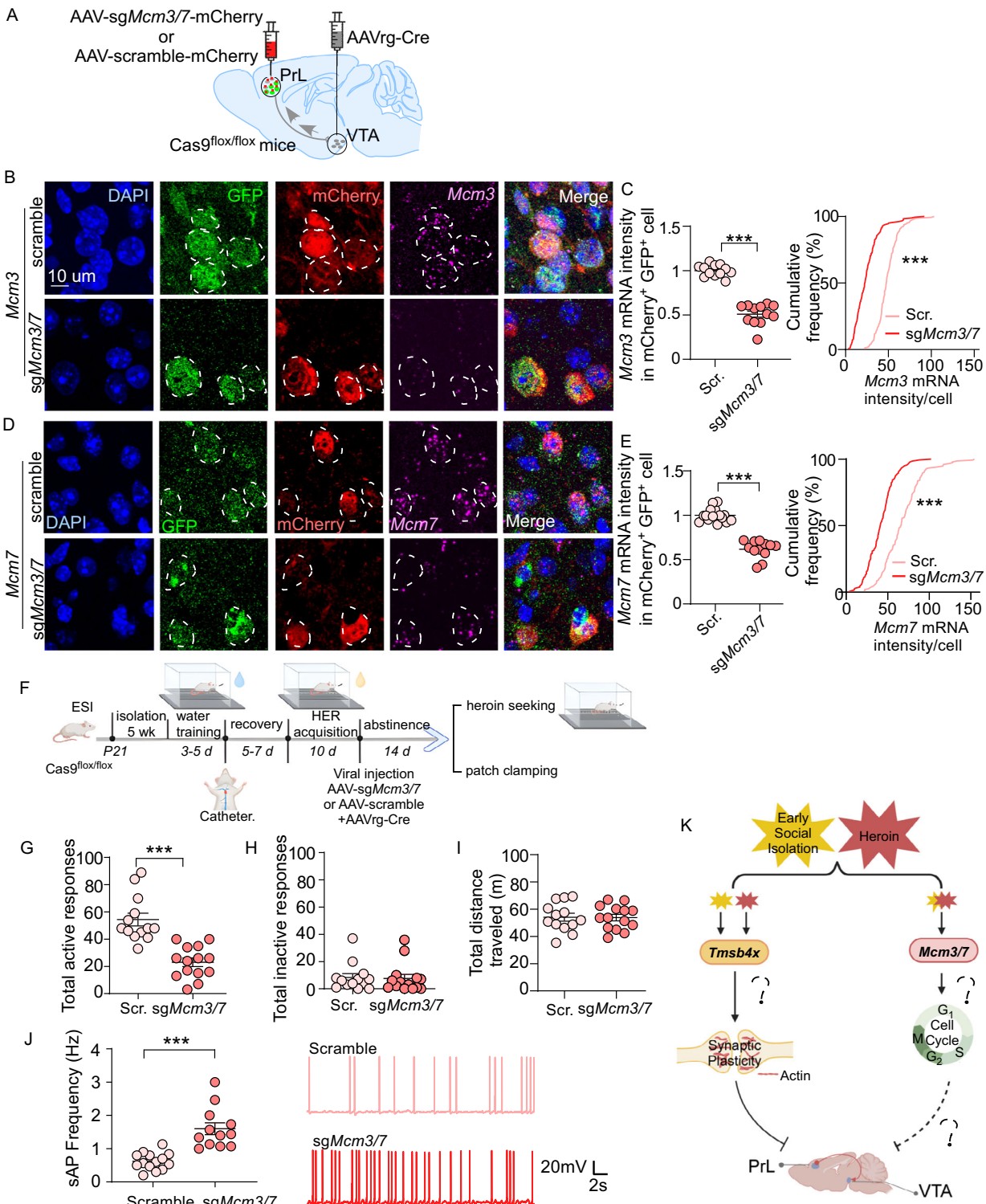

**Mcm3 and Mcm7 mediate PrL->VTA circuit vulnerability induced by the interaction between ESI stress and heroin abstinence.** Stress and opioid exposure can both induce DNA damage[109–112]. Our data suggest that ESI stress and heroin abstinence interactively affect the expression of genes involved in cell cycle and DNA damage responses (Fig. 6). In post-mitotic neurons, cell cycle re-entry and DNA damage response are closely linked. For example, the DNA damage response can trigger cell cycle re-entry, accompanied by activation of cell cycle markers and DNA synthesis, leading to neuronal apoptosis[83,84,113]. On the other hand, cell cycle activation can be part of DNA repair that

prevents immediate neuron death[114]. We identified two key players among the genes regulating cell cycle and DNA damage response: Mcm3 and Mcm7. They are part of the MCM protein complex, which gates DNA replication licensing to ensure the replication occurs just once per cell cycle[80,81]. Intriguingly, Mcm7 gene expression was reduced in the postmortem PFC tissue from opioid users[115], along with altered pathways related to DNA damage and repair[111,115]. We also found that heroin abstinence reduced the expression of Mcm3 and Mcm7 (Fig. 6E–I), which may potentially limit DNA replication to restrict cell cycle re-entry-induced cellular disturbance and preserve neuronal

**Fig. 7 | Knockdown of *Mcm3* and *Mcm7* in PrL->VTA projection attenuates heroin-seeking behavior in ESI mice. A** Schematic of virus injection in Cas9[flox/flox] mice. **B–E** Representative images of *Mcm3* (**B**) and *Mcm7* (**D**) mRNA expression in sg*Mcm3/7* or scramble groups. Blue: DAPI; Green: AAVrg-Cre-mediated Cas9-GFP expression in PrL->VTA projecting neurons; Red: mCherry; Pink: *Mcm3* or *Mcm7*. Quantification of the fluorescent intensity of *Mcm3* (**C**, sg*Mcm3/7*: *n* = total 183 cells [for cumulative frequency] from 12 images [for mRNA intensity] from 5 mice; scramble: *n* = total 162 cells [for cumulative intensity] from 12 images [for mRNA intensity] from 5 mice; $t_{22}$ = 12.48, *p* < 0.0001) and *Mcm7* (**E**, sg*Mcm3/7*: *n* = total 181 cells [for cumulative frequency] from 15 images [for mRNA intensity] from 5 mice; scramble: *n* = total 153 cells [for cumulative frequency] from 12 images [for mRNA intensity] from 5 mice; $t_{25}$ = 11.26, *p* < 0.0001) mRNA in mCherry[+]GFP[+] cells (unpaired two-tailed *t*-test and two-sample KS test, *p* < 0.0001). **F** Experimental timeline for water training, heroin (HER) self-administration, virus injection,

behavioral testing, and electrophysiology in ESI Cas9[flox/flox] mice. Mean numbers of total active responses (**G**, n = 13 [scr.]–14 [sg*Mcm3/7*] mice/group, unpaired two-tailed *t*-test, $t_{25}$ = 5.66, *p* < 0.0001) and total inactive responses (**H**) during heroin seeking. (**I**) Mean numbers of total distance traveled (m) during locomotion test (*n* = 13 [scr.]–14 [sg*Mcm3/7*] mice/group). **J** Top: Graph showing the firing frequency of sAP in PrL->VTA projection between scramble and sg*Mcm3/7* groups (*n* = 14 [scr.]–12 [sg*Mcm3/7*] cells from 4–6 mice/group, unpaired two-tailed *t*-test, $t_{24}$ = 5.44, *p* < 0.0001). Bottom: representative sAP traces. **K** Schematic of potential mechanisms of how early social isolation stress and heroin abstinence together induce the hypofunction of PrL->VTA projection to potentiate heroin seeking. Data are expressed as mean ± SEM, ***p* < 0.001. **F**, **K** created in BioRender. Wang, Z. (2025) https://BioRender.com/6kfkeg5. Source data are provided as a Source data file.

function. Compared to the GH HER group, *Mcm3* and *Mcm7* expression levels in PrL->VTA projection are increased in ESI mice that underwent heroin abstinence (Fig. 6E–I). This suggests prior ESI stress experience renders the *Mcm3* and *Mcm7* responses to heroin abstinence impaired, which can potentially lead to cell cycle re-entry and cellular disruption and result in dysfunction of PrL->VTA projection. However, MCMs also function beyond replication, such as regulating transcription, chromatin remodeling[116–118], and the DNA damage checkpoint[82,116]. In addition, genes that modulate the cell cycle have well-established roles in regulating other processes, such as neuronal and synaptic functions[119,120]. How these functions are involved in the ESI-potentiated heroin-seeking needs further exploration, as does the differential regulation of *Mcm3* and *Mcm7* gene expression by heroin abstinence alone and by the interaction between ESI stress and heroin.

One limitation of the current study is the unclear connection between *Tmsb4x* and *Mcm3/7* in existing network databases (e.g., STRING). This lack of clarity makes it challenging to understand how *Tmsb4x* and *Mcm3/7* interact (at transcriptional and/or translational levels) to contribute to PrL->VTA circuit vulnerability. Further study is needed to explore the potential interaction between *Tmsb4x* and *Mcm3/7* in detail.

### Early life stress increases vulnerability to opioid relapse and sex differences

Our data suggest that ESI stress creates vulnerable brain circuitry (i.e., PrL->VTA projection) by changing the transcriptional profile and neuronal activity to induce increased risk for opioid relapse, i.e., potentiated heroin-seeking behavior. This is consistent with our previous study[21] and other labs reporting that social isolation stress increases seeking or extinction resistance for several drugs in different behavioral procedures[20,121–123]. Interestingly, contradictory results are reported in studies using short-term early life stressors: limited bedding and nesting (postnatal days 2–9) promote resilience to opioid addiction-related phenotypes in male rats[124]. These results highlight that early life stress is associated with opioid addiction in a stressor type- and timing-dependent manner.

It is noteworthy that animal studies showed that social isolation stress or opioid exposure can induce sex-specific functional and transcriptional changes[31,33,35,125,126]. For example, ESI induces PrL hypoactivity in male and female mice[35,126], including diminished spontaneous firing rate and excitatory synaptic transmission in PrL pyramidal neurons, but only induces VTA hypoactivity in female mice. Long-term abstinence from remifentanil SA reduces the excitability and excitatory synaptic transmission in PrL layer V pyramidal neurons in female mice[31]. While we found that the PrL->VTA projection hypoactivity (reduced spontaneous firing rate and excitatory synaptic transmission) was in combined sexes. These studies highlight the importance of studying specific cell types (e.g., VTA dopamine neurons[35,126] or overall PrL layer V pyramidal neurons[31] versus neurons within the PrL->VTA projection). Furthermore, adolescent social isolation stress induces

sex-specific transcriptional changes in PFC and VTA[33,35], while we identified several universal key genes (*Tmsb4x*, *Mcm3*, and *Mcm7*) mediating the transcriptional effect of social isolation and heroin abstinence on PrL->VTA projection in both sexes (Fig. S11). However, it is hard to rule out the sex-specific functional and transcriptional changes contributing to the hypofunction of PrL->VTA projection induced by adolescent social isolation stress and opioid exposure. These potential sex-dependent changes need further investigation.

## Conclusion

Taken together, our data indicate that ESI stress aggravates the disrupted neuronal firing and excitatory synaptic transmission in the PrL->VTA projection after heroin abstinence. Chemogenetic activation of PrL->VTA projection attenuates heroin-seeking behavior in ESI mice while inhibition of this projection in GH mice potentiates heroin seeking. Furthermore, ESI stress and heroin abstinence collectively alter the transcriptional profile within the PrL->VTA projection to induce disrupted neuronal function, contributing to stress-potentiated heroin seeking. These results provide neural and molecular mechanisms for early life stress-induced vulnerability to opioid addiction.

## Methods
### Animals
C57BL/6J mice (stock number 000664), B6;129S4-*Gt(ROSA)26 Sor*[tm9(EGFP/Rpl10a)Amc]/J (GFP-L10a) mice (stock number 024750), B6;129S6-*Gt(ROSA)26Sor*[tm2(CAG-NuTRAP)Evdr]/J (NuTRAP) mice (stock number 029899), and B6J.129(Cg)-*Gt(ROSA)26Sor*[tm1.1(CAG-cas9*,-EGFP)Fezh]/J mice (Cas9[flox/flox] mice, stock number 026179) were purchased from Jackson lab (Bar Harbor, ME, USA). Both male and female mice were used in this study. All animals were maintained on a 12-h light/dark cycle (lights off at 11:00 A.M. and light on at 11:00 P.M.) with ad libitum access to water and food chow. All animal experiments were performed under the approval of the Institutional Animal Care and Use Committee (IACUC). All animals were maintained according to the National Institutes of Health guidelines in Association for Assessment and Accreditation of Laboratory Animal Care-accredited facilities.

### Early social isolation stress
ESI was initiated immediately after weaning, from postnatal day 21 (P21) to P60 (5 weeks), based on previous publications[21,32,35]. Control mice were group-housed (GH) at 3–5 mice per cage.

### Drug
Heroin hydrochloride was a gift from the NIDA drug supply program and was dissolved in sterile saline. Heroin solutions (0.1 mg/mL) were prepared weekly. Pump durations were adjusted daily according to the animals' body weights (0.05 mg/kg/infusion). Compound 21 (C21, Tocris, Cat. No. 5548, 1 mg/mL) was prepared by dissolving in sterile saline and then diluted with sterile saline before use. Thymosin β4 (Tβ4) (Gibco™ 14014100UG, 20 μg/mL) was prepared by dissolving in sterile PBS.

## Self-administration test chambers

The experimental chambers were set up as previously described[70,127,128] with modifications. In brief, 16 standard Med Associates Inc. (St. Albans, VT) chambers were used, each with two nose-poke holes equipped with infrared monitoring. Two stimulus lights were mounted within each nose-poke hole, with a house light in the center back wall of the test chamber. All chambers were housed in sound-attenuating boxes and controlled through a Med Associates interface.

## Jugular catheterization surgery

Mice were deeply anesthetized with 100 mg/kg ketamine and 5 mg/kg xylazine, then implanted with chronic indwelling jugular catheters as previously described[21,112,127–129] with modifications. The catheter was inserted into the right jugular vein and sutured. It was then threaded subcutaneously over the shoulder blade and connected to the harness (Instech, Plymouth Meeting, PA, USA). Following surgery, catheters were flushed daily with 0.05 mL of heparinized saline to preserve catheter patency. Before the initiation of SA training and after the last session, each animal received an IV infusion of ketamine hydrochloride (1 mg/mL in 0.05 mL), and the behavioral response was observed to verify catheter patency. A decrease in muscle tone served as a behavioral indicator of patency. Jugular catheterization surgeries were performed on both heroin- and non-heroin-exposed mice to ensure a consistent surgical experience across groups.

## Self-administration (SA)

All SA experiments occurred in the standard SA test chambers described above. The saline or heroin SA procedures were conducted as previously described[21,70,112,128] with modifications. Briefly, all mice were subjected to 3–5 days of water training (one session per day, with 12–16 h of water restriction before training) before jugular catheterization surgery. During water training, the mice were trained to make nose-poke responses to get water delivered directly into the active hole (20 μL). The fixed ratio (FR) for water training increased from a fixed ratio 1 (FR1) to FR3 across the 3 days. Mice were given 20 min of water access after each water training session. Jugular catheterization surgery was performed after water training, and mice were then allowed to recover for 5–7 days and were assigned to the acquisition of heroin or saline SA. Mice were subjected to daily 3-h SA training, during which responses to the active alternative snout-poke hole resulted in IV infusions of heroin (0.05 mg/kg/infusion) or sterile saline according to an FR1 schedule of reinforcement, which was increased daily to FR3 (FR1 on Day 1, FR2 on Day 2, and FR3 on Day 3 and thereafter) and maintained at this FR for the remaining SA protocol (one training session/day). The active snout-poke hole was different from the one used for water training to avoid the extinction of water responding or generalization to the heroin nose-poke. Infusions were accompanied by a 5-s illumination of the stimulus light inside the active snout-poke hole, followed by a 5-s timeout period, during which the house light was extinguished. Responses to the inactive hole resulted in no programmed consequences. The following criteria for the acquisition of operant responding were adopted from publications with modifications[21]: mice will be included if they maintain stable responding with (1) less than 30% deviation from the mean of the total number of infusions earned in the last three consecutive sessions, (2) at least 65% responding on the reinforced nose-poke, and (3) a minimum of 5 reinforcers per session.

## Heroin-seeking test

After heroin acquisition, animals underwent forced abstinence for 14 days, a period that showed high extinction responding to heroin[130], mimicking real-life situations where environmental cues precipitate relapse behavior following an extended period of abstinence. Then, mice were placed back in the same chambers for a 1-h context- and cue-induced seeking test (referred to as the heroin-seeking test). During the heroin-seeking test, active responses produced discrete cues previously paired with drug delivery, but heroin was not available.

## Locomotion test

Mouse locomotion was tested for 15 min in an open-field transparent plastic apparatus (70 × 50 × 35 cm) immediately following the heroin-seeking test. The floor of the chamber was marked with two evenly spaced vertical and three horizontal black lines, dividing the floor into a 3 × 4 grid of 12 equal rectangles. At the start of the test, each mouse was gently placed in one corner of the apparatus, with its head facing the corner. Locomotion was monitored using a video surveillance system (iSpy, Western Australia, Australia). Ambulatory activity was quantified by manually counting the number of line crossings, defined as the mouse moving all four paws from one rectangle into another. In different batches of experiments, the total distance traveled within 15 min was analyzed using AnyMaze software (Stoelting Co.).

## Retrograde tracer injection

C57BL/6J mice were deeply anesthetized with 100 mg/kg ketamine and 5 mg/kg xylazine and placed on a stereotaxic apparatus (RWD Instruments, San Diego, CA, USA). Body temperature was maintained with a heating pad throughout the surgery. To target the PrL->VTA projection, 400 nL of Cholera Toxin Subunit B (CTB) 488 or CTB 555 (Invitrogen, CA, USA) was bilaterally injected into the VTA (AP: −3.4, ML: ±0.46, DV: −4.2). Injections were performed at a rate of 50 nL/min using a micropump (RWD Instruments). The needle was held in the target position for 10 min after the injection was completed.

## Designer receptors exclusively activated by designer drugs (DREADDs)

All viruses were directly purchased from the vendor. C57BL/6J mice were deeply anesthetized with 100 mg/kg ketamine and 5 mg/kg xylazine and placed on a stereotaxic apparatus (RWD Instruments). Body temperature was maintained with a heating pad throughout the surgery. To target the PrL-> VTA projection, 400 nL of AAV2-CMV-eGFP-Cre (Addgene, #105545-AAV2) or DIO-hM4Di-mCherry (BrainVTA, #PT 1143) was bilaterally injected into the PrL (AP: +2.4, ML: ±0.3, DV: −1.8). Five hundred nanoliters of a retrograde pAAV-hSyn-DIO-hM3Dq-mCherry (Addgene, #44361-AAVrg) or AAVrg-Cre (Addgene, #55636-AAVrg) were bilaterally injected into the VTA (AP: −3.4, ML: ±0.46, DV: −4.2) (Fig. 2B). Injections were performed at a rate of 50 nL/min using a micropump (RWD Instruments). The needle was set in the target position for 10 min after the injection was completed. Heroin-seeking test was performed 2 weeks after surgery, following the acute (30 min before heroin seeking) intraperitoneal (i.p.) injection of C21 (Tocris, Cat. No. 5548) (1 mg/kg body weight, in a working solution with saline of 0.1 mg/mL[57]). Animals were counterbalanced based on the average SA performance for the last 4 sessions and assigned to receive C21 injections.

## Thymosin β4 microinjection

Following heroin SA, mice were subject to a 14-day abstinence, during which bilateral guide cannulae were implanted. Surgery was performed according to previous publications[90,128,131]. Mice were anesthetized with 100 mg/kg ketamine and 5 mg/kg xylazine and positioned in a stereotaxic frame (RWD Instruments). A double guide cannula with a dummy (RWD Instruments) was implanted into the PrL (AP: +2.4, ML: ±0.3, DV: −1.3) and fixed with dental cement. A microinjector with a 0.5-mm protrusion below the guide cannula aiming at the PrL (−1.8 mm) was used for infusion. Mice were counterbalanced according to SA performance during the last 4 sessions and assigned to receive vehicle or thymosin β4 (20 μg/mL, 1 μL/hemisphere) infusion (0.5 μL/min using 100-μL Hamilton syringes) controlled by a syringe pump (RWD Instruments) during abstinence days 8–14[75] (one infusion daily). After infusion, the needle was kept at the injection site for another 5 min to allow complete diffusion.

## Electrophysiology

The whole-cell patch-clamp recording technique was used to measure AP and synaptic transmission in layer V pyramidal neurons of prefrontal cortical slices, as previously described[132–134]. Mice were euthanized with 1–3% isoflurane inhalation followed by decapitation. Brain tissue was rapidly removed, iced, and cut into 300 μm coronal slices using a vibratome (VT1000s, Leica, Wetzlar, Germany). Brain slices were allowed to recover in oxygenated (95% $O_2$ and 5% $CO_2$) ACSF (130 mM NaCl, 26 mM $NaHCO_3$, 3 mM KCl, 5 mM $MgCl_2$, 1.25 mM $NaH_2PO_4$, 1 mM $CaCl_2$, 10 mM glucose, pH 7.60, 300 mOsm) for 40 min in a 33 °C water bath and then kept at room temperature for 30 min prior to the start of recording. The slices were positioned in a perfusion chamber attached to the fixed stage of an upright microscope (Scientifica, East Sussex, UK) and submerged in continuously flowing oxygenated ACSF (1 mL/min). PrL pyramidal neurons projecting to the VTA (labeled with CTB [retrograde tracing in Figs. 1 and 5], or double-labeled with GFP and mCherry [Figs. 2 and 7], or labeled with mCherry [Fig. 3]), were visualized using a 40× water-immersed lens with an upright microscope (Scientifica, East Sussex, UK). Patch pipette (3–6 MΩ) was pulled from 1.5 mm borosilicate glass capillaries using a micropipette puller (P-1000, Sutter Instrument, CA, USA) and filled with internal solution containing 124 mM K-gluconate, 1 mM $MgCl_2$, 6 mM KCl, 5 mM EGTA, 10 mM HEPES, 0.5 mM $CaCl_2$, 0.5 mM $Na_2GTP$, 3 mM $Na_2ATP$, 12 mM phosphocreatine, pH 7.2–7.3, 265–270 mOsm.

For the recording of sEPSCs, the membrane potential was maintained at −70 mV, and bicuculline (20 μM) was added to isolate the excitatory transmission. APs were recorded in current clamp mode, and slices were bathed in a modified ACSF containing 130 mM NaCl, 26 mM $NaHCO_3$, 3.5 mM KCl, 0.5 mM $MgCl_2$, 1.25 mM $NaH_2PO_4$, 1 mM $CaCl_2$, 10 mM glucose, pH 7.60, 300 mOsm. To record spontaneous APs (sAP), a small depolarizing current was applied to adjust the inter-spike potential to −60 to −65 mV. To elicit current-evoked APs (eAP), a series of 600-ms current (0–200 pA, 20 pA increment) was injected. To minimize experimental variations, only cells with comparable membrane capacitances and series resistance that changed by less than 10% were included[132–134]. Data were acquired using a Multiclamp 700B amplifier (Molecular Devices, CA, USA) and digitized with a DigiData 1550B data acquisition board (Molecular Devices). Cells were allowed to stabilize for at least 2 min before recording. The recorded signals were digitized at 5 kHz, filtered at 1 kHz, and collected using Clampex 11.0 data acquisition system (Molecular Devices). Recorded sEPSC data were analyzed with MiniAnalysis (Synaptosoft, NJ, USA). Two to six cells per mouse were used for the electrophysiology study (cell and animal numbers for each experiment were provided in the figure legend).

## Immunohistochemistry and analysis

To validate the effect of DREADDs on the PrL->VTA projection activity, we performed immunofluorescent staining for c-Fos. Animals were euthanized 30 min after C21 or saline injection. Mice were deeply anesthetized with ketamine/xylazine and transcardially perfused with sterile-filtered 0.01 M PBS, pH 7.4 (phosphate-buffered saline), followed by sterile-filtered 4% paraformaldehyde (PFA) in 0.01 M PBS, pH 7.4, at 4 °C. Whole brains were immediately harvested and post-fixed in 4% PFA at 4 °C for 24 h and then immersed in 30% sucrose in 0.01 M PBS (pH 7.4) at 4 °C for cryoprotection. Coronal sections encompassing the PrL and VTA were cut at a thickness of 40 μm using a vibratome (Leica VT1000s, Buffalo Grove, IL, USA) and then stored at −20 °C until use. For immunostaining, brain sections were rinsed in 0.01 M PBS (pH 7.4) 3 times, then blocked with 3% normal serum (NS) in PBS containing 0.3% Triton-X for 1 h at room temperature. Sections were then incubated with primary antibody c-Fos (1:500, Abcam, ab190289, Cambridge, MA, USA/ Cell Signaling, #2250S), CaMKII (1:200, Abcam, ab22609), Mcm3 (1:50, Santa Cruz, sc-390480) or Mcm7 (1:50, Santa Cruz, sc-9966) diluted in 3% NS in PBS with 0.1% saponin overnight at 4 °C. On the second day, after three PBS washes, sections were

incubated with anti-Alexa Fluor 350-conjugated, anti-Alexa Fluor 488-conjugated, or anti-Alexa Fluor 555-conjugated secondary antibody at 1:200 (Thermo Fisher, Waltham, MA, USA) for 2 h at room temperature. Coverslips were attached to the slides with mounting media (Vector Lab, Burlingame, CA, USA). Images were taken on a Nikon Ti2 Eclipse microscope. c-Fos-positive nuclei were measured using ImageJ software according to previous publications[21]. Mcm3 and Mcm7 protein intensities in the PrL->VTA projecting neurons were measured by identifying CTB retrogradely labeled target cells using ImageJ. Two to three slices (images) per mouse were used for the immunohistochemistry study (image and animal numbers for each experiment were provided in the figure legend).

## Projection-specific molecular profiling by retro-translating ribosome affinity purification (retro-TRAP)

**Surgical procedures.** GH and ESI GFP-L10a mice were subjected to 10 days of saline or heroin SA. One day after the last SA session, mice were deeply anesthetized with 100 mg/kg ketamine and 5 mg/kg xylazine and placed on a stereotaxic apparatus (RWD instruments, CA, USA). Their body temperature was maintained with a heating pad. To target PrL -> VTA neurons, 300 nL of retrograde AAV carrying Cre recombinase (Addgene, #55636-AAVrg) was bilaterally injected into the VTA (AP: −3.4, ML: ±0.46, DV: −4.2). Injections were performed at a rate of 50 nL/min using a micropump (KD Scientific, MA, USA). The needle was kept in the target position for 10 min after the injection was completed.

**TRAP.** Thirteen days after viral surgery, TRAP was performed as described[135–137] with modifications. In brief, mice were anesthetized with 3% isoflurane inhalation and decapitated. Brains were quickly removed and sliced with a pre-chilled 1 mm coronal brain matrix (RWD Instruments). Brain slices were placed in ice-cold 0.01 M PBS, and brain punches containing PrL were immediately homogenized in ice-cold supplemented homogenization buffer (50 mM Tris pH 7.4, 100 mM KCl, 12 mM $MgCl_2$, and 1% NP-40 in RNase-free water) in a glass homogenizer. Homogenates were centrifuged at 10,000 RPM for 10 min at 4 °C to pellet nuclei and large cell debris. Two anti-GFP antibodies, HtzGFP-19F7 and HtzGFP-19C8 (Memorial Sloan Kettering Cancer Center, New York, NY), were added to the cell lysate and incubated overnight at 4 °C. Anti-GFP-coated Dyna magnetic beads (Invitrogen, CA, USA) were washed with homogenization buffer. The beads were then added to the lysate, and the mixture was incubated overnight at 4 °C. Polysome-RNA complexes bound to the anti-GFP-coated streptavidin magnetic beads were separated from the supernatant by a magnet; RNA was isolated using the Absolutely RNA Nanoprep kit (Agilent, #400753, USA). RNA was purified with in-column DNase digestion. RNA quantity and quality were determined by the Tapestation RNA 6000 Pico Chip (Agilent Technologies, Santa Clara, CA) and Qubit (Invitrogen, CA, USA).

## Quantitative real-time PCR

TRAP RNA was purified as described above. First-strand complementary DNA (cDNA) was synthesized using PrimeScript™ Reverse Transcriptase (2680B, TaKaRa, CA, USA). Then the cDNA was amplified in the StepOnePlus™ Real-Time PCR System (Applied Biosystems, MA, USA) using PowerUp™ SYBR™ Green Master Mix (A25742, Applied Biosystems) and sequence-specific primers. *Gapdh* was used as a housekeeping gene for quantitation. Fold change was determined by: Fold change = $2^{-\Delta(\Delta CT)}$, where ΔCT = CT (gene of interest)−CT (*Gapdh*), and Δ(ΔCT) = ΔCT (treated group)−ΔCT (control). Primer sequences used were provided in Table S1.

## RNA-sequencing (RNA-seq) and bioinformatic analysis

TRAP RNA from each group (GH SAL, $n = 3$ mice; ESI SAL, $n = 3$ mice; GH HER, $n = 3$ mice; ESI HER, $n = 3$ mice) was purified as described above. The RNA-sequencing libraries were constructed using the

NEBNext Ultra II directional RNA library prep kit for Illumina via the ribosomal RNA depletion method (New England BioLabs, Ipswich, MA). 150 bp paired-end reads were generated on the NextSeq2000 platform (Illumina, CA, USA) at the Genome Sequencing Core at the University of Kansas.

Bioinformatics analyses were conducted based on previous publications with modifications[35,132,133]. In brief, after quality checks using FastQC (all samples passed quality check), RNA-seq reads were first trimmed using Cutadapt to remove the 3′ end adapters and trailing sequences, and then aligned to the mouse GRCM38-mm10 RefSeq mRNAs using HISAT2 with default parameters. In GRCm38-mm10, all of the 20,423 component sequences were applied in mapping and annotation. Alignments with a mapping score <10 were discarded using SAMtools. A matrix of mapped fragments per UCSC RefSeq known annotated gene was generated with FeatureCounts. Genes annotated as rRNA were discarded.

Full threshold-free differential expression was detected by DESeq2 with default negative binomial generalized linear models[138]: the $p$-value was calculated by the default Wald test, and the adjusted $p$-value was calculated by the Benjamini-Hochberg test (representing the False Discovery Rate [FDR]) for multiple corrections. The cutoff for identifying DEGs is adjusted $p$ (i.e., FDR) < 0.1 unless otherwise specified. To detect the interaction effect between stress (GH or ESI stress) and drug (saline or heroin) on gene expression, DESeq2 coupled with LRT analysis[138,139] was used. In this method, the full model (representing the overall effect) was "stress + drug + stress:drug," and the reduced model (representing stress and drug independent effects) was "stress + drug"; the interaction between stress and drug (stress:drug) represents whether the effect of one variable on its on-site occurrence depends on the level of another, and can be assessed by comparing the full model ("stress + drug + stress:drug") to the reduced model ("stress + drug"). For each gene, the interaction fold change (FC) can be approximated as: $FC \approx \frac{\frac{ESI-HER}{ESI-SAL}}{\frac{GH-HER}{GH-SAL}} = \frac{\frac{ESI-HER}{GH-HER}}{\frac{ESI-SAL}{GH-SAL}}$. This approximation can be interpreted in two equivalent ways: it represents (i) either the ratio of heroin-induced changes in the ESI group relative to those in the GH group, or (ii) the ratio of ESI-induced changes in the HER group relative to those in the saline (SAL) group. Thus, this approximation of FC ultimately reflects the difference between ESI- and heroin-induced changes (i.e., interaction effects). MetaData for DESeq2 analysis were provided in Table S2.

Heatmaps and volcano plots were generated using gplots and ggplot2. Venn diagrams were generated using InteractiVenn[140]. Statistical significance of the overlap between two groups of DEGs was calculated by the hypergeometric probability formula using the online tool "Statistical significance of the overlap between two groups of genes" (http://nemates.org/MA/progs/overlap_stats.html). A full threshold-free RRHO map was generated using the RRHO package (version 1.38.0) as previously described[141]. Functional protein classification analyses were undertaken using PANTHER. Enrichment analyses of DEGs were conducted using gene sets derived from the Biological Process Ontology from DAVID using a cutoff $p < 0.05$. Synaptic GO analysis was conducted using gene sets from SynGO. Cytoscape software was used to visualize the gene interaction and network for the top 40 DEGs (rank = −log10 ($p$ value) × |fold change|) that were affected by the interaction effect between ESI stress and heroin abstinence. The CytoHubba application with the MCC (Maximal Clique Centrality) algorithm[142] was applied to identify hub genes. STRING Enrichment[143] and EnrichmentMap[144] plugins were applied to identify node genes and analyze pathway enrichment using a cutoff FDR < 0.05.

### Conditional knockdown of *Mcm3* and *Mcm7* by CRISPR-Cas9 system

sgRNAs were designed using online CRISPR tools (CRISPOR): spsgRNA (5′-CCACACAGACCACACAGCTG-3′) targeting exon 3 of *Mcm3*, spsgRNA (5′-AAACTTACCTGGAAGCCCAC-3′) targeting exon 8 of *Mcm7*, and scramble sgRNA (5′-GTTCAGGATCACGTTACCGC-3′). To investigate the specificity of CRISPR-Cas9-mediated knockdown of *Mcm3* and *Mcm7*, Cas-OFFinder[145] was used to predict the off-target binding sites of *Mcm3* and *Mcm7* gRNA in the mouse (*Mus musculus* [mm10]) reference genome with the following parameters: Pam type = SpCas9 from Streptococcus pyogenes: 5′-NGG-3′, mismatch number = 3, DNA bulge size = 0, and RNA bulge size = 0. Neuro-2A (N2A) cells were cultured in Dulbecco's modified Eagle's medium (Thermo Fisher, 11965092) supplemented with 10% fetal bovine serum (Thermo Fisher, 26140079). The U6-spsgRNA (*Mcm3*) and spsgRNA (*Mcm7*) cassettes were cloned into the spCas9-NLS-Flag-P2A-Puro-T2A-EGFP backbone from a commercial resource (BrainVTA Co., Ltd). Cells were transfected with FuGENE HD transfection reagent (Promega, E2311). For selection, cells were treated with puromycin (2 μg/mL) for 3 days. Genomic DNA of puromycin-selected cells was extracted using QIAamp DNA mini kit (Qiagen #51304) and PCR amplification was performed using the following primers: *Mcm3* on-target primer forward, 5′-TACAGGCTGTCGTTGTCGTC-3′; reverse, 5′-AGTGAACA CTGCGAACGACT-3′; *Shroom3* off-target forward, 5′-TGACTTTC CCGTAGGGTTGC-3′; reverse, 5′- TGGGGACACTTAGGATGCAC-3′; *Celf2* off-target forward, 5′-GTTGGGGGAATCTCGGTGTT-3′; reverse, 5′-CAACACGCATGCTCACACTC-3′; *Mcm7* on-target primer forward, 5′-TATGAGTCCTGCAGCTGGTTA-3′; reverse, 5′-CCTCCCTCCTAAGTC TCTTGTCC-3′; *Ano2* off-target forward, 5′-GCTGCACCTGAGAAAG GGAT-3′; reverse, 5′-AATAGCCAGCAACTCCCACC-3′; *Ptgfrn* off-target forward, 5′-CAGCCTCAACCTGCACACTA-3′; reverse, 5′-GGCCCAC AGGAAGCTGTAAT-3 (Table S3). PCR products were digested with T7 endonuclease and analyzed by DNA gel electrophoresis. Cleavage efficiency was calculated as: cleavage efficiency = $1 - (1 - fraction\ cleaved)^{1/2}$, where fraction cleaved = sum of cleaved-band intensities/ (sum of the cleaved and parental band intensities). Two hundred nanoliters of rAAV-U6-spsgRNA(*Mcm3*)-U6-spsgRNA(*Mcm7*)-pCBh-mCherry-WPRE or control virus rAAV-U6-spsgRNA(scramble)-U6-spsgRNA(scramble)-pCBh-mCherry-WPRE (BrainVTA Co., Ltd) was bilaterally injected into the PrL (AP: +2.4, ML: ±0.3, DV: −1.8). Additionally, 400 nL of retrograde AAV carrying Cre recombinase (Addgene, #55636-AAVrg) was bilaterally injected into the VTA (AP: −3.4, ML: ±0.46, DV: −4.2) of Cas9$^{flox/flox}$ mice to initiate Cas9 expression in PrL->VTA projecting neurons. After behavior tests, we measured the *Mcm3* and *Mcm7* knockdown efficiency by RNAscope.

### RNAscope

RNAscope® Multiplex Fluorescent Assay v2 from Advanced Cell Diagnostics (ACDBio, Abingdon, United Kingdom) and dyes from Akoya Biosciences were used. Brains collected from Cas9$^{flox/flox}$ mice were sliced on a cryostat at 15 μm and mounted directly onto Superfrost Plus slides (Fisher Scientific). The tissue was fixed in 4% formaldehyde for 15 min and then dehydrated in ethanol. Endogenous peroxidase was blocked with hydrogen peroxide for 10 min at room temperature, and then target retrieval and proteolysis using RNAscope® Pretreatment Reagents (ACD 322381) were performed. The tissue was incubated for 2 h at 40 °C using RNAscope® Probes - Mm-*Mcm3* (ACD 563201) and -Mm-*Mcm7* (ACD 1332061) for mouse tissue. After hybridization, probes were labeled with Akoya Opal fluorophore reagents. Sections were counterstained with 4, 6-diamidino-2-phenylindole (DAPI) and coverslipped with mounting media (Vector Lab, Burlingame, CA, USA). Images were acquired using a Leica Laser Scanning Confocal Upright Microscope.

### Statistical analysis

All data were analyzed using SPSS software (IBM Corp., Armonk, NY, USA) or GraphPad (GraphPad Software, San Diego, CA), and were represented as the mean ± SEM, with $p < 0.05$ indicating significance. Multi-factor repeated ANOVA was performed on the number of infusions and nose-poke responses during acquisition for heroin SA, as

well as the evoked AP elicited by different injected currents. All other comparisons of dependent variables were analyzed by two-way ANOVA for parametric data. Post hoc analysis was performed by Tukey's comparisons. Comparisons between two groups were analyzed with an unpaired, two-tailed *t*-test. For data that did not pass the normality test, the Kruskal–Wallis test (for multiple comparisons) or the Mann–Whitney test (for two-group comparisons) was used. The number of animals in each group is provided in the figure legends. Statistical details are provided in Supplementary Data 1.

### Reporting summary

Further information on research design is available in the Nature Portfolio Reporting Summary linked to this article.

## Data availability

The data supporting the findings of this study are available in the article and in its online supplementary material. Genomic data are available through the GEO database (GSE293281). Source data are provided with this paper.

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

## Acknowledgements

The authors would like to thank the NIDA drug supply program for providing controlled substances. The authors would also like to thank Dr. Stuart J. Macdonald (University of Kansas) for the guidance on RNA-seq data analysis. The authors would also like to thank Mr. Benjamin J Macdonald, Ms. Arushi Garg, and Ms. Esther Anderson for their help with animal care and data analysis. The authors also thank Dr. Liqin Zhao's lab (University of Kansas) for providing the N2A cell line. Z.J.W. is supported by NIH grants DA050908 and DA056804. Y.W., S.Y., and L.C. are supported by the Sobek-Zhang Foundation. Research reported in this study was made possible in part by the services of the Genome Sequencing Core lab at the University of Kansas. This lab is supported by the National Institute of General Medical Sciences (NIGMS) of the National Institutes of Health under award number P30GM145499.

## Author contributions

Y.W. and S.Y.: performed experiments, analyzed data, and drafted the manuscript. F.Y., L.C., A.S., M.M. and W.W.: performed experiments and analyzed data. Z.J.W.: designed experiments, supervised the project, and wrote the manuscript.

## Competing interests

The authors declare no competing interests.
