## [Transparent Peer Review file · Nature Communications]

Prelimbic cortex to ventral tegmental area projection regulates early social isolation stress-potentiated heroin seeking in mice

Corresponding Author: Dr Zi-Jun Wang

Version 0:

Reviewer comments:

Reviewer #1

(Remarks to the Author)

This extensive report examined a new circuit that contributed to the early life stress-induced vulnerability of heroin relapse. It was found that ESI stress exacerbated heroin-induced neuronal dysfunction in the prefrontal cortex (PrL) to ventral tegmental area (VTA) projecting neurons. Then, authors activated the PrL->VTA projection which led to the attenuation of ESI-potentiated heroin seeking, as well as normalized neuronal firing and excitatory synaptic transmission in these neurons. At the genetic level, ESI stress and heroin abstinence affected the expression of genes regulating morphogenesis and metabolic processes, and they identified *Tmsb4x* as one of the hub genes mediating ESI-potentiated heroin seeking.

This is a well-designed and executed study. The results are clear, behavioral data robust and the conclusion was supported by the results. The manuscript is overall well-written.

I did not find major issues overall. It is a solid set of data that should have broad interest to the field of addiction neurobiology.

Reviewer #2

(Remarks to the Author)

In this set of studies, Wang and colleagues explore the neural mechanisms underlying enhanced heroin-seeking behavior after early social isolation (ESI) stress in mice. In particular, they explore projections from prefrontal (PL) cortex to VTA in mice that have undergone heroin self-administration, abstinence, and then a seeking relapse test. Using electrophysiological techniques, they find that VTA-projecting PL neurons have reduced excitatory activity after heroin abstinence and ESI alone and the effect is magnified with both ESI and heroin. Chemogenetic inhibition of VTA-projecting PL neurons reduces heroin seeking in ESI and non-ESI mice and restores excitability in the cells. Chemogenetic activation increases heroin seeking in non-ESI mice and reduces cellular excitability. RNA-seq analysis of VTA-projecting PL cells (using retro-TRAP), as compared to all PL, revealed several differentially expressed genes (DEGs) related to heroin abstinence or ESI, including 11 overlapping genes for heroin and ESI. One of the top-ranked hub genes was *Tmsb4x*, which encodes thymosin beta-4. Microinjection of recombinant thymosin beta-4 in PL of mice during heroin abstinence reduced subsequent seeking. Further analysis of genes involved in interaction between ESI and heroin revealed many DEGs, including hub genes *Mcm3* and *Mcm7*, which are involved in gating DNA replication and repair. Conditional knockdown of *MCM3* and *MCM7* in VTA-projecting PL neurons via CRISPR-Cas9 caused reduced heroin seeking in ESI mice and restored cellular excitability. Overall, this is a very well-written manuscript with a set of experiments that is well executed, using innovative tools and multi-level approach. The authors have conducted a very thorough examination of their research question, with well explained methods and results. The graphics and figures are useful and easy to follow.

My primary suggestion for edits would be to include a better description in the abstract and/or introduction concerning the experiments and results. As it's written now, it's not clear what experiments were done until the reader gets to the Methods and Results, which are explained very well. A better abstract will orient the reader, attract more readers, and highlight the innovative tools used and major findings. It would be helpful to include more detail for the parts related to dysfunction (cellular

excitability), activation (chemogenetic), genes (describe specificity to VTA-projecting PL cells, RNA-seq and retro-TRAP; say what Tmsb4x and Mcm genes encode), include manipulations for these two gene sets (thymosin injections, conditional knockdown). Although there are word count limits, some details are important.

A minor comment is that it is not clear what unit is used for locomotion on the figures. The methods indicate lines crossed or distance traveled for 15 min, but different figures have pretty different averages (different y-axes), so it's not clear if this is due to differences in measures or group means.

Reviewer #3

(Remarks to the Author)

Reviewer comments for 'Prelimbic cortex to ventral tegmental area projection regulates early social isolation stress potentiated heroin seeking' by Wang et al.

The authors previously established a preclinical model of stress-potentiated heroin seeking using mice. Early life stress resulted in a significant increase in heroin seeking as determined by active lever responses after forced abstinence. In the current study, the authors sought to provide more mechanistic support of their behavioral observations by examining the physiological responses of prelimbic (PrL to VTA) circuits that are involved in heroin seeking behavior. Using a combination of genetically modified mice and viral tools, the authors used DREADDs to activate or inactivate the PrL to VTA projection and further demonstrated that manipulation of this circuit can rescue stress-induced potentiation of heroin seeking. Further molecular analysis was performed on PrL VTA projection neurons by RNA sequencing using the RNA-TRAP methodology. The authors identified transcripts that were enriched in the PrL VTA circuits as opposed to PrL bulk RNA seq; and further stratified the data following stress +/-heroin SA. From this analysis, they identified Tmsb4x, as a key transcript associated with ESI stress-potentiated PrL->VTA projection hypofunction and heroin seeking. The reproducibility of the ESI-potentiated heroin seeking is a key strength of this study and provides strong rationale for investigating the mechanisms of such behavioral consequences. Further information provided in the methodology would greatly improve the quality of the manuscript.

Additional details are required for many sections of the methods to ensure reproducibility of results:

- The methodological description of SA would benefit from further details regarding when the mice transition from FR1 to FR3 such as which day and for how many sessions. Are these methods the same for each experiment?
- For the description of the locomotion test, the authors reference that the total number of lines crossed was calculated. How many lines were in the chamber? Because such tests can often be scored as time in the center vs the peripheral section of the locomotor chamber, clarification of this point is needed. Are the lines referring to inner versus out quadrants of the chamber or a series of beam breaks to calculate ambulation?
- The description of the DREADD experiments references AddGene constructs. When reviewing the catalog numbers provided in the manuscript, it is not clear if the authors procured plasmid constructs and packaged the plasmids into viruses themselves, or obtained viruses directly from Addgene. Thus, clarification of these methods is needed.
- For the injection of thymosin B4, the authors state that microinjections were performed during abstinence days 8-14. The authors should clarify how many injections of thymosin B4 the animals received, and on which days.
- Additional methodological details are needed to provide the reader with the number of cells per animal that were used for electrophysiological recordings to obtain the results. The same comment goes for the immunohistochemical analyses.
- For the description of the RNA sequencing methods, the authors state that the cut off for DEGs was set at an adjusted p value of less than 0.1. However, the authors should clarify if all the genes classified as significant at this level were also, a minimum, significant for an unadjusted p value of <0.05. The authors describe a method of analysis of DESeq2 with LRT analysis. References from the literature for this approach, if available, would provide more context for the reader.
- The statistical description does not mention whether test for normality were performed prior to applying parametric statistical analyses. The authors should perform normality tests and then select the appropriate statistical test.

General:

- The use of the word 'sacrifice' throughout the manuscript should be eliminated, and instead replaced with 'euthanized'.
- Typos at lines 342 (linear); 380 (remove 'and'); 406 (text says that authors used probes targeting probes and is redundant); 449 (mice THAT underwent)...There are several grammatical English language errors that should be corrected throughout the manuscript prior to publication.
- Adding a description to the supplemental excel files to detail what data is being presented and further details regarding metadata is needed.
- In regards to the RNA seq analysis of PrL bulk RNA seq versus PrL VTA enriched gene expression, the authors states in the methods that they used an adjusted p-value of 0.1 that was not significant as their cutoff for DEGs. However, they report '1374 down-regulated and 3525 up-regulated genes. This represents about 25% of the transcriptome and may be expected. However, it may also allow for false positive DEGs, as the less stringent p-value was used. The authors should re-analyze their dataset and use a significant adjusted p-value of <0.05.
- For the gene expression analysis in Figure 4, it would be helpful for the reader to know the comparisons that are being made for the volcano plots, gene ontology, and venn diagrams. It may be assumed that the comparisons are ESI vs GH; and Her vs Sal. However, this should be included in the figure legend at a minimum.

-For the gene expression analyses performed in the study, were the non-heroin exposed mice also catheterized? Did they also undergo saline SA? This is a very important control, as some of the gene expression changes observed to overlap between ESI (no heroin) and heroin could simply be due to the surgical stress and catheterization.

-In figure 4, it is difficult to see the gene that is being measured in each graph. Putting the gene examined at the top of the graph or in bold would be helpful, as some gene names overlap on two graphs and it's hard to tell what the name is referring to.

- Throughout the manuscript there is inconsistent nomenclature referring to gene names. Authors should check guidelines regarding punctuation and italics at all gene name mentions.
- The organization of figure 4 could be improved. The panels go Left Right then Up down without a particular consistency. Figure 7 also is similarly inconsistent and difficult to follow.
- How was the overlap analysis performed in Fig. 44?
- For Figure 6, I am not able to evaluate the data based on the information presented. Further information is required to understand exactly what comparisons were made and are presented in the Figure as well as the accompanying datasets.

Version 1:

Reviewer comments:

Reviewer #1

(Remarks to the Author)

I congratulate authors for this nice contribution.

Reviewer #2

(Remarks to the Author)

The authors have addressed all my concerns.

Reviewer #3

(Remarks to the Author)

The authors have submitted a revised manuscript that thoroughly and rigorously addressed all critiques raised by this reviewer. I have no additional comments for the authors.

REVIEWER COMMENTS

We sincerely appreciate the editor and reviewers for their positive and constructive comments. We have incorporated ALL of the suggestions, including: adding details to the abstract and introduction, correcting typos and language errors, providing further details on experiment design, re-analyzing some of the RNA-seq datasets, adding one more supplemental table, and reorganizing some figure panels. All revisions are marked in Blue in the manuscript. The point-to-point response to the comments is listed below.

Reviewer #1 (Remarks to the Author):

This extensive report examined a new circuit that contributed to the early life stress-induced vulnerability of heroin relapse. It was found that ESI stress exacerbated heroin-induced neuronal dysfunction in the prelimbic cortex (PrL) to ventral tegmental area (VTA) projecting neurons. Then, authors activated the PrL->VTA projection which led to the attenuation of ESI-potentiated heroin seeking, as well as normalized neuronal firing and excitatory synaptic transmission in these neurons. At the genetic level, ESI stress and heroin abstinence affected the expression of genes regulating morphogenesis and metabolic processes, and they identified *Tmsb4x* as one of the hub genes mediating ESI-potentiated heroin seeking.

This is a well-designed and executed study. The results are clear, behavioral data robust and the conclusion was supported by the results. The manuscript is overall well-written.

I did not find major issues overall. It is a solid set of data that should have broad interest to the field of addiction neurobiology.

We greatly thank the reviewer for these encouraging comments.

Reviewer #2 (Remarks to the Author):

In this set of studies, Wang and colleagues explore the neural mechanisms underlying enhanced heroin-seeking behavior after early social isolation (ESI) stress in mice. In particular, they explore projections from prelimbic (PL) cortex to VTA in mice that have undergone heroin self-administration, abstinence, and then a seeking relapse test. Using electrophysiological techniques, they find that VTA-projecting PL neurons have reduced excitatory activity after heroin abstinence and ESI alone and the effect is magnified with both ESI and heroin. Chemogenetic inhibition of VTA-projecting PL neurons reduces heroin seeking in ESI and non-ESI mice and restores excitability in the cells. Chemogenetic activation increases heroin seeking in non-ESI mice and reduces cellular excitability. RNA-seq analysis of VTA-projecting PL cells (using retro-TRAP), as compared to all PL, revealed several differentially expressed genes (DEGs) related to heroin abstinence or ESI, including 11 overlapping genes for heroin and ESI. One of the

top-ranked hub genes was *Tmsb4x*, which encodes thymosin beta-4. Microinjection of recombinant thymosin beta-4 in PL of mice during heroin abstinence reduced subsequent seeking. Further analysis of genes involved in interaction between ESI and heroin revealed many DEGs, including hub genes *Mcm3* and *Mcm7*, which are involved in gating DNA replication and repair. Conditional knockdown of *MCM3* and *MCM7* in VTA-projecting PL neurons via CRISP-Cas9 caused reduced heroin seeking in ESI mice and restored cellular excitability. Overall, this is a very well-written manuscript with a set of experiments that is well executed, using innovative tools and multi-level approach. The authors have conducted a very thorough examination of their research question, with well explained methods and results. The graphics and figures are useful and easy to follow.

My primary suggestion for edits would be to include a better description in the abstract and/or introduction concerning the experiments and results. As it's written now, it's not clear what experiments were done until the reader gets to the Methods and Results, which are explained very well. A better abstract will orient the reader, attract more readers, and highlight the innovative tools used and major findings. It would be helpful to include more detail for the parts related to dysfunction (cellular excitability), activation (chemogenetic), genes (describe specificity to VTA-projecting PL cells, RNA-seq and retro-TRAP; say what *Tmsb4x* and *Mcm* genes encode), include manipulations for these two gene sets (thymosin injections, conditional knockdown). Although there are word count limits, some details are important.

We greatly appreciate the reviewer's suggestion. We have added details about the experiments and results to both the abstract and introduction. As the reviewer mentioned, the Abstract has a word limit (200), so more details are included in the introduction with an additional paragraph at the end. These added changes are listed below:

Abstract:

“.....Chemogenetic activation of PrL->VTA projection.....”

“.....alongside normalized neuronal firing and excitatory synaptic transmission in these neurons.”

“RNA-seq revealed that ESI stress and heroin abstinence.....”

“.....with *Tmsb4x* (encoding thymosin β 4) as one of the hub genes.....”

“.....with *Mcm3* and *Mcm7* (encoding minichromosome maintenance protein 3 and 7) as hub genes. Thymosin β 4 infusion in PrL or CRISPR-Cas9-mediated *Mcm3/7* knockdown in PrL->VTA projection attenuated ESI-potentiated heroin seeking and neuronal hypofunction.”

Introduction:

“To do so, we used the forced abstinence model and focused on the PrL rather than the IL, which is involved in extinction learning⁵⁰. Using whole-cell patch-clamp recording,

we identified reduced neuronal firing and excitatory synaptic transmission in PrL->VTA projections after early social isolation stress and heroin exposure. Chemogenetic activation of PrL->VTA projections alleviated ESI-potentiated heroin seeking and restored neuronal function in this circuit. Finally, using translating ribosome affinity purification (TRAP)-coupled with RNA-seq, we found the transcriptional alterations within this projection that drive early life stress-potentiated heroin seeking, with *Tmsb4x* (encoding thymosin β 4), *Mcm3* and *Mcm7* (encoding minichromosome maintenance proteins 3 and 7) identified as hub genes. Further manipulation—via thymosin β 4 infusions in the PrL or CRISPR-Cas9-mediated conditional knockdown of *Mcm3* and *Mcm7* in the PrL->VTA projection—validated their key roles in mediating ESI-induced susceptibility to heroin.”

A minor comment is that it is not clear what unit is used for locomotion on the figures. The methods indicate lines crossed or distance traveled for 15 min, but different figures have pretty different averages (different y-axes), so it's not clear if this is due to differences in measures or group means.

We apologize for the confusion. Initially, we did locomotion by measuring the line crossing. Later, we acquired the Anymaze software to track locomotion by measuring the traveled distance. This is why we had two different measurements of locomotion, and the numbers for lines crossed or distance traveled are widely different. We have now added the exact locomotion measurement (either the number of line crossings or total distance traveled) in each figure.

Reviewer #3 (Remarks to the Author):

Reviewer comments for 'Prelimbic cortex to ventral tegmental area projection regulates early social isolation stress potentiated heroin seeking' by Wang et al.

The authors previously established a preclinical model of stress-potentiated heroin seeking using mice. Early life stress resulted in a significant increase in heroin seeking as determined by active lever responses after forced abstinence. In the current study, the authors sought to provide more mechanistic support of their behavioral observations by examining the physiological responses of prelimbic (PrL to VTA) circuits that are involved in heroin seeking behavior. Using a combination of genetically modified mice and viral tools, the authors used DREADDs to activate or inactivate the PrL to VTA projection and further demonstrated that manipulation of this circuit can rescue stress-induced potentiation of heroin seeking. Further molecular analysis was performed on PrL Δ VTA projection neurons by RNA sequencing using the RNA-TRAP methodology. The authors identified transcripts that were enriched in the PrL Δ VTA circuits as opposed to PrL bulk RNA seq; and further stratified the data following stress +/-heroin SA. From this analysis, they identified *Tmsb4x*, as a key transcript associated with ESI stress-

potentiated PrL->VTA projection hypofunction and heroin seeking. The reproducibility of the ESI-potentiated heroin seeking is a key strength of this study and provides strong rationale for investigating the mechanisms of such behavioral consequences. Further information provided in the methodology would greatly improve the quality of the manuscript.

Additional details are required for many sections of the methods to ensure reproducibility of results:

- The methodological description of SA would benefit from further details regarding when the mice transition from FR1 to FR3 such as which day and for how many sessions. Are these methods the same for each experiment?

We apologize for the missing information. We have added the details to the **Self-administration** method part as below. These methods are consistent across all experiments.

“.....which was increased daily to FR3 (FR1 on Day 1, FR2 on Day 2, and FR3 on Day 3 and thereafter) and maintained at this FR for the remaining SA protocol (one training session/day).”

- For the description of the locomotion test, the authors reference that the total number of lines crossed was calculated. How many lines were in the chamber? Because such tests can often be scored as time in the center vs the peripheral section of the locomotor chamber, clarification of this point is needed. Are the lines referring to inner versus out quadrants of the chamber or a series of beam breaks to calculate ambulation?

We are sorry for the lack of information. The locomotor chamber was gridded with 2 vertical lines and 3 horizontal lines, creating a total of 12 distinct squares. Our measurement of locomotion was based on the number of “beam breaks” as the animal moved within the chamber rather than on the time spent in specific zones. Therefore, each line crossing corresponded to a “beam break” event. As we don’t have a sensor to detect the actual “beam break”, the number of line crossings is used to evaluate locomotion. Later, we acquired the Anymaze software to evaluate locomotion using the total distance traveled. Therefore, we have some batches of locomotion experiments measured by “number of line crossings” and others measured by “total distance traveled”, which are reflected in the figures now. We have revised the **locomotion test** method as below:

“The floor of the chamber was marked with two evenly spaced vertical and three horizontal black lines, dividing the floor into a 3 × 4 grid of 12 equal rectangles. At the start of the test, each mouse was gently placed in one corner of the apparatus, with its head facing the corner. Locomotion was monitored using a video surveillance system (iSpy, Western Australia, Australia). Ambulatory activity was quantified by manually counting the number of line crossings, defined as the mouse moving all four paws from

one rectangle into another. In different batches of experiments, the total distance traveled within 15 min was analyzed using AnyMaze software (Stoelting Co.).”

- The description of the DREADD experiments references AddGene constructs. When reviewing the catalog numbers provided in the manuscript, it is not clear if the authors procured plasmid constructs and packaged the plasmids into viruses themselves, or obtained viruses directly from Addgene. Thus, clarification of these methods is needed. We apologize for this missing information. All viruses are directly purchased from the vendor. We have included the virus catalog number in the Methods and added the following sentence at the beginning of the DREADDs methods: “All viruses were directly purchased from the vendor.”

- For the injection of thymosin B4, the authors state that microinjections were performed during abstinence days 8-14. The authors should clarify how many injections of thymosin B4 the animals received, and on which days.

We have added the microinjection details in the **Thymosin β 4 Microinjection** method part as below:

“.....controlled by a syringe pump (RWD Instruments) during abstinence days 8–14⁶⁰ (one infusion daily).....”

- Additional methodological details are needed to provide the reader with the number of cells per animal that were used for electrophysiological recordings to obtain the results. The same comment goes for the immunohistochemical analyses.

We have added the cell/slice numbers per animal for e-phys and IHC in the **Electrophysiology and Immunohistochemistry and Analysis** method parts as below:

“....Two to six cells per mouse were used for the electrophysiology study (cell and animal numbers for each experiment were provided in the figure legend).”

“...Two to three slices (images) per mouse were used for the immunohistochemistry study (image and animal numbers for each experiment were provided in the figure legend).”

- For the description of the RNA sequencing methods, the authors state that the cut off for DEGs was set at an adjusted p value of less than 0.1. However, the authors should clarify if all the genes classified as significant at this level were also, a minimum, significant for an unadjusted p value of <0.05. The authors describe a method of analysis of DESeq2 with LRT analysis. References from the literature for this approach, if available, would provide more context for the reader.

We apologize for the confusion about the p-value and adjusted p-value. In DESeq2 analysis, the Benjamini-Hochberg method was used to adjust p-values and control the False Discovery Rate (FDR). In our data sets, the adjusted p value represents the FDR, which is a more stringent measurement compared to the unadjusted p value. We have clarified this concept in the **RNA sequencing (RNA-seq) and bioinformatic analysis**

method parts as below. When the adjusted p value is 0.1, the corresponding unadjusted p value ranges from 0.0027 to 0.0011. For the LRT analysis, we have added two references to provide more information for the readers.

“.....the p-value was calculated by the default Wald test, and the adjusted p-value was calculated by the Benjamini-Hochberg test (representing the False Discovery Rate [FDR]) for multiple corrections. The cutoff for identifying differentially expressed genes (DEGs) is adjusted p (i.e., FDR) < 0.1 unless otherwise specified.”

References for LRT:

67. Love, M.I., Huber, W., and Anders, S. (2014). Moderated estimation of fold change and dispersion for RNA-seq data with DESeq2. *Genome Biol* 15, 550. 10.1186/s13059-014-0550-8.

68. Love, M.I., Simon Anders, and Huber, W. (2025). Analyzing RNA-seq data with DESeq2.

<https://bioconductor.org/packages/devel/bioc/vignettes/DESeq2/inst/doc/DESeq2.html>.

- The statistical description does not mention whether test for normality were performed prior to applying parametric statistical analyses. The authors should perform normality tests and then select the appropriate statistical test.

We greatly appreciate this suggestion from the reviewer. We have now performed the normality test on our datasets. For the majority of our 4-group comparisons, data from at least three groups passed the normality test, and the Q-Q plot generally aligned with a straight line. For these datasets, we applied the ANOVA test. We chose ANOVA because it is robust to violations of the normality assumption when testing hypotheses about means, especially when the non-normal group, as in our datasets, isn't extremely skewed or has severe outliers. This approach is supported by the publication *Feir-Walsh, B. J., & Toothaker, L. E. [1974]. An empirical comparison of the ANOVA F-test, normal scores test and Kruskal-Wallis test under violation of assumptions. Educational and Psychological Measurement, 34[4], 789–799.*

Several datasets related to inactive responses (during drug seeking) did not pass the normality test. We have updated their statistical analysis accordingly, though these changes in statistical analysis led to the same conclusions as before. For example, in Fig. 2F (inactive responses), we now used the Kruskal-Wallis test, which continues to show a significant difference among groups. A post-hoc Mann-Whitney test further confirms a difference in inactive responses between GH-Veh and GH-C21 groups. Similarly, for Fig. 5D and Fig. 7H (inactive responses), we now used the Mann-Whitney test, which still showed no difference between the two groups, consistent with our prior findings. We have updated the statistical methods for Fig. 2F, Fig. 5D and Fig. 7H in the supplementary statistical table. We also updated the Methods for **Statistical analysis** as below to reflect these changes:

“For data that did not pass the normality test, the Kruskal-Wallis test (for multiple comparisons) or the Mann-Whitney test (for two-group comparisons) was used.”

General:

-The use of the word ‘sacrifice’ throughout the manuscript should be eliminated, and instead replaced with ‘euthanized’.

We have changed all “sacrifice” into “euthanize”.

-Typos at lines 342 (linear); 380 (remove ‘and’); 406 (text says that authors used probes targeting probes and is redundant); 449 (mice THAT underwent)...There are several grammatical English language errors that should be corrected throughout the manuscript prior to publication.

We're grateful for the reviewer's careful evaluation of our manuscript. We've corrected these and other language-related errors throughout the text.

-Adding a description to the supplemental excel files to detail what data is being presented and further details regarding metadata is needed.

We thank the reviewer for this comment. We have added the description of the Excel sheet on the Title line (first line) of each supplementary Excel table to show what data is presented. We also added the metadata Excel sheet used for the DEseq2 analysis as a new supplemental Excel file (Table S2).

-In regards to the RNA seq analysis of PrL bulk RNA seq versus PrL→VTA enriched gene expression, the authors states in the methods that they used an adjusted p-value of 0.1 that was not significant as their cutoff for DEGs. However, they report ‘1374 down-regulated and 3525 up-regulated genes. This represents about 25% of the transcriptome and may be expected. However, it may also allow for false positive DEGs, as the less stringent p-value was used. The authors should re-analyze their dataset and use a significant adjusted p-value of <0.05.

We deeply apologize for the confusion here. Initially, we used the cutoff of adjusted p-value (FDR) <0.05 and $|\text{Log}_2\text{FC}|>1$ for bulk tissue RNA vs PrL→VTA TRAP mRNA analysis, due to the largely distinct transcriptional profiles between the two groups. We have specified this in the results section. However, we completely agree with the reviewer that 25% of the transcriptome may still include false-positive results. Therefore, we re-analyzed the data using more stringent criteria: $\text{FDR}<0.01$ and $|\text{Log}_2\text{FC}|>1$, as suggested. With these stricter parameters, we found that 3293 genes showed higher expression while 1283 genes showed lower expression in PrL→VTA TRAP mRNA compared to bulk RNA. Even with the $\text{FDR}<0.01$ criterion, the overlap between our current DEGs and the published DEGs (Murugan et al., 2017) remains significant. These results reinforce that we isolated the PrL→VTA TRAP mRNA, which contains distinct transcriptional profiles. We have added this new analysis to the Results section

and the end of supplementary figure 8. If the reviewer suggests using even more stringent criteria, we could analyze this data again.

Results:

“.....we found that PrL->VTA projection has a distinct transcriptional profile compared to total bulk RNA, with 1374 down-regulated and 3525 up-regulated genes (Table S5) using the cutoff of adjusted p value (FDR) <0.05 and |Log2FC|>1; or 1283 down-regulated and 3293 up-regulated genes when using a more stringent criterion FDR<0.01 and |Log2FC|>1.”

-For the gene expression analysis in Figure 4, it would be helpful for the reader to know the comparisons that are being made for the volcano plots, gene ontology, and venn diagrams. It may be assumed that the comparisons are ESI vs GH; and Her vs Sal. However, this should be included in the figure legend at a minimum.

We apologize for the confusion. We have added the comparison in the Results and Figure legends for Fig. 4 as below:

Results:

“.....We identified that 195 DEGs were impacted by ESI stress (DEG^{ESI}: ESI vs GH, Figure 4A, Table S6).....”

“.....We also found that 374 DEGs were impacted by abstinence from heroin self-administration (DEG^{HER}: HER vs SAL, Figure 4B, Table S7).....”

Figure legend:

“.....differentially expressed genes (DEGs) induced by ESI stress (A, DEG^{ESI}: ESI vs GH) or heroin (HER) abstinence (B, DEG^{HER}: HER vs SAL) in PrL->VTA projection.....”

“Gene Ontology (GO) pathway enrichment analysis for DEGs induced by ESI stress (C, ESI vs GH) or HER abstinence (D, HER vs SAL).”

“Venn diagram showing the overlapping between DEGs caused by ESI stress (DEG^{ESI}: ESI vs GH) and HER abstinence (DEG^{HER}: HER vs SAL).”

-For the gene expression analyses performed in the study, were the non-heroin exposed mice also catheterized? Did they also undergo saline SA? This is a very important control, as some of the gene expression changes observed to overlap between ESI (no heroin) and heroin could simply be due to the surgical stress and catheterization.

Yes, the non-heroin mice in both GH and ESI groups were catheterized and subjected to saline self-administration training so that these procedures-induced changes in gene expression can be cancelled off. We have now emphasized this information as below:

Jugular Catheterization Surgery method: “.....Jugular catheterization surgeries were performed on both heroin- and non-heroin-exposed mice to ensure a consistent surgical experience across groups.”

Results: “....GH or ESI GFP-L10a mice were subjected to saline or heroin SA (Figure S9A-E)....”

-In figure 4, it is difficult to see the gene that is being measured in each graph. Putting the gene examined at the top of the graph or in bold would be helpful, as some gene names overlap on two graphs and it's hard to tell what the name is referring to.

We thank the reviewer for raising this issue. We have put the gene names on top of each panel to make it clear.

-Throughout the manuscript there is inconsistent nomenclature referring to gene names. Authors should check guidelines regarding punctuation and italics at all gene name mentions.

We thank the reviewer for pointing this out. We've now standardized all mouse gene symbols to comply with convention: italicized, with the first letter capitalized and the remaining letters lowercase.

-The organization of figure 4 could be improved. The panels go Left \diamond Right then Up \diamond down without a particular consistency. Figure 7 also is similarly inconsistent and difficult to follow.

We are sorry for the confusion. We have reorganized Fig. 4 and Fig. 7 for easy understanding.

- How was the overlap analysis performed in Fig. 44?

We calculated the probability of overlap using the online tool "Statistical significance of the overlap between two groups of genes"

(http://nemates.org/MA/progs/overlap_stats.html). This information is now added to

RNA sequencing (RNA-seq) and bioinformatic analysis method. The calculation is based on the hypergeometric distribution as below: $hypergeometric(x; s, M, N) =$

$\frac{\binom{M}{x} \binom{N-M}{s-x}}{\binom{N}{s}}$, where x is the overlapping genes, s is the number of genes in the new list, M is

the number of genes in the population that are also present in new list and N is the set of all genes, and the brackets indicate the binomial coefficient: $\binom{a}{b} = \frac{a!}{b!(a-b)!}$. A P -value

is obtained from the Cumulative Distribution Function (CDF) by integrating (summing) the probability distribution from one extremity of the distribution to the value of x :

$Hypergeometric(x; s, M, N) \begin{cases} \sum_{i=0}^x hypergemetric(i; s, M, N) & x \leq \bar{x} \\ \sum_{i=x}^s hypergemetric(i; s, M, N) & x > \bar{x} \end{cases}$ where $\bar{x} = s \frac{M}{N}$ is the expected value of the hypergeometric distribution.

-For Figure 6, I am not able to evaluate the data based on the information presented. Further information is required to understand exactly what comparisons were made and are presented in the Figure as well as the accompanying datasets.

We apologize for the confusion. Figure 6 evaluates the interaction effects between early social isolation (ESI) and heroin (HER) on gene expression in our RNA-seq datasets (MetaData in Supplemental Table 2). These interaction effects (or interaction DEGs) were identified with DEseq2 using a likelihood ratio test (LRT), in which a full model including the interaction effect (stress + drug + stress:drug) was compared to a reduced model without it (stress + drug). For each gene, the interaction fold change (FC) can be approximated

$$\text{as: } FC \approx \frac{\frac{ESI-HER}{GH-HER}}{\frac{ESI-SAL}{GH-SAL}} = \frac{ESI-HER}{ESI-SAL} \cdot \frac{GH-SAL}{GH-HER} .$$

This approximation can be interpreted in two equivalent ways: (i) as the ratio of heroin-induced changes in the ESI group relative to those in the GH group, or (ii) as the ratio of ESI-induced changes in the HER group relative to those in the saline (SAL) group. Thus, this approximation of FC ultimately reflects the difference between ESI- and heroin-induced changes (i.e., interaction effects).

If ESI and HER cause identical changes in both direction and magnitude (e.g., Fig. R1a), then $FC = 1$, and Log_2FC for the interaction effect would be zero. Conversely, deviations from 0 in Log_2FC indicate that the effects of ESI and HER differ (in direction, magnitude, or both [e.g., Fig. R1 b-f]), suggesting there is an interaction effect. In other words, the two factors (in our case, ESI and HER) are interfering with one another to change the outcome (in our case, gene expression).

In our datasets, increased Log_2FC suggests that HER-induced changes are greater in the GH group compared to the ESI group, or that ESI-induced changes are greater in the HER group compared to the SAL group. Conversely, decreased Log_2FC implies that HER-induced changes are smaller in the GH group compared to the ESI group, or that ESI-induced changes are smaller in the HER group compared to the SAL group.

The approximate formula for FC has now been included in both the **RNA sequencing (RNA-seq) and bioinformatic analysis** method part and the Fig. 6 legend to facilitate the understanding of the figure. We emphasized that the above FC calculation

Fig. R1. Schematic depiction of data scenarios without and with interaction effect (IE). (a) Group 0 (blue) and 1 (red) both have a positive effect for treatment high compared to low and a positive group effect, but no IE. (b) As in (a), but with an additional positive IE. (c) Negative IE between group and treatment. (d) No treatment effect for group 0. The treatment effect for group 1 is entirely represented by the IE. (e) Both groups display a positive treatment effect and there is no group effect in the treatment category low, only in high, i.e. an IE is present. (f) Negative IE between group and treatment, but no line crossing as in (c). [Duda J.C. et al., 2023]

is an approximation intended to help readers conceptually understand the interaction effect, as DESeq2 derives the actual FC from model coefficient estimations obtained through generalized linear model fitting of our RNA-seq data, rather than from direct sample means.